# Distance-informed Neural Processes

**Aishwarya Venkataramanan**
Computer Vision Group
Friedrich Schiller University Jena, Germany
`aishwarya.venkataramanan@uni-jena.de`

**Joachim Denzler**
Computer Vision Group
Friedrich Schiller University Jena, Germany
`joachim.denzler@uni-jena.de`

## Abstract

We propose the Distance-informed Neural Process (DNP), a novel variant of Neural Processes that improves uncertainty estimation by combining global and distance-aware local latent structures. Standard Neural Processes (NPs) often rely on a global latent variable and struggle with uncertainty calibration and capturing local data dependencies. DNP addresses these limitations by introducing a global latent variable to model task-level variations and a local latent variable to capture input similarity within a distance-preserving latent space. This is achieved through bi-Lipschitz regularization, which bounds distortions in input relationships and encourages the preservation of relative distances in the latent space. This modeling approach allows DNP to produce better-calibrated uncertainty estimates and more effectively distinguish in- from out-of-distribution data. Empirical results demonstrate that DNP achieves strong predictive performance and improved uncertainty calibration across regression and classification tasks.

## 1 Introduction

Deep neural networks have achieved remarkable success in fields such as computer vision [5, 60], natural language processing [57, 42], and reinforcement learning [9]. However, they often produce overconfident and unreliable predictions, particularly when encountering data that fall outside the support of the training set [23, 58, 17, 1, 59]. Stochastic models such as Gaussian Processes (GPs) [66] address this limitation by defining a prior over functions using kernel functions, which encode the assumption that similar inputs produce similar outputs. Kernel functions measure similarity using a distance metric, assigning higher scores to inputs closer to the training data. Consequently, when the model encounters data points that are far from the training set, the predictive distribution reverts to the prior. In many cases, this prior has a high variance, indicating a high uncertainty about the output. However, the practicality of vanilla GPs is limited by their high computational cost, scaling cubically with the size of the training data and by their lack of flexibility in high-dimensional problems.

This paved way to the development of Neural Processes (NPs) [11, 12], that leverage meta-learning principles [20] to learn a distribution over functions from observed data. Unlike traditional kernel-based methods, NPs employ neural encoders to infer a global latent variable that summarizes the underlying function. The NP then conditions its predictions on the inputs and the global latent variable, facilitating rapid adaptation to new functions. However, the reliance on a single global latent variable limits representational expressiveness, and standard NPs often produce uncertainty estimates that are not well calibrated [14, 25, 63].

To address these limitations, we propose the *Distance-informed Neural Process* (DNP), which integrates both global and local priors for better uncertainty estimation and generalization. The global prior functions like standard NPs, capturing task-specific patterns. The local prior models distance relationships between observed data and test inputs through a latent space, introducing an inductive bias similar to GPs. However, neural networks used to parameterize these latent spaces can warp the input geometry causing similar points to be mapped far apart and unrelated points

39th Conference on Neural Information Processing Systems (NeurIPS 2025).

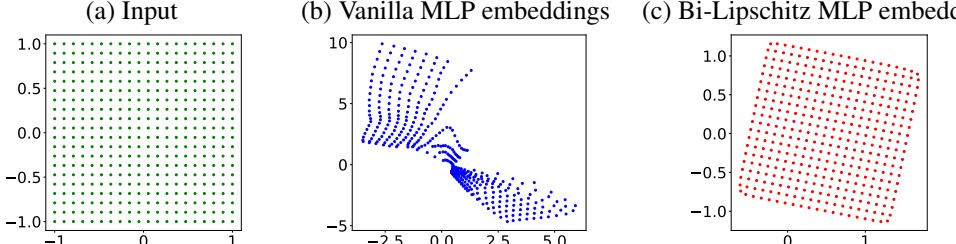

Figure 1: Visualization of input data and learned representations on a 2D synthetic data. (a) shows the original data. (b) shows representations from a 3-layer vanilla MLP, where the input structure is distorted. (c) shows representations from the MLP with bi-Lipschitz regularization, which better preserves the input structure and relative distances, important for computing reliable input similarity.

to appear close together in the learned representation [36, 55, 69]. To mitigate this, we add a bi-Lipschitz regularization on the local latent network's weight matrices [68]. This constrains each layer's singular values within a fixed range, which minimizes distortions and encourages the latent space to approximately preserve input distances [36]. The distance-preserving latent space is then used to define a local prior that encodes data similarity. As shown in Fig. 1, the latent representations learned by a vanilla neural network distorts input geometry, while bi-Lipschitz regularization helps preserve the input structure by minimizing such distortions[1]. By conditioning on both global and distance-aware local latent variables, DNP yields better-calibrated uncertainty estimates and improved distinction between in- and out-of-distribution (OOD) data. We demonstrate its effectiveness in regression and classification through experiments on synthetic and real-world datasets. Code is available: `https://github.com/cvjena/DNP.git`.

## 2   Background

Let $\mathcal{D} = (\mathbf{x}_i, \mathbf{y}_i)_{i=1}^{N}$ represent a dataset consisting of input-output pairs, where $\mathbf{x}_i \in \mathbb{R}^{d_x}$ is an input and $\mathbf{y}_i \in \mathbb{R}^{d_y}$ is the corresponding output. NPs define a family of conditional distributions over functions $f \in \mathcal{F}$, mapping $\mathbf{x}$ to $\mathbf{y}$. The dataset is split into a context set $(\mathbf{x}_C, \mathbf{y}_C) = (\mathbf{x}_i, \mathbf{y}_i)_{i=1}^{M}$, where $M \leq N$ is an arbitrary subset of $\mathcal{D}$, and a target set $(\mathbf{x}_T, \mathbf{y}_T) = (\mathbf{x}_i, \mathbf{y}_i)_{i=1}^{N}$, which includes all data points. The model uses the context set to create a summary that conditions how it predicts the target outputs $\mathbf{y}_T$ from the target inputs $\mathbf{x}_T$. Different variants of NPs differ in how this context-based conditioning is performed and how the predictive distribution is defined.

The **Conditional Neural Process** (CNP) [11] models the conditional distribution as $p(\mathbf{y}_T|\mathbf{x}_C, \mathbf{y}_C, \mathbf{x}_T) = p(\mathbf{y}_T|\mathbf{r}_C, \mathbf{x}_T)$, where $\mathbf{r}_C = \text{Agg}((\mathbf{x}_c, \mathbf{y}_c)_{c \in C})$ is a permutation-invariant summary (e.g. mean) of the context points. Each $(\mathbf{x}_c, \mathbf{y}_c)$ is first encoded by a neural network, after which $\mathbf{r}_C$ and $\mathbf{x}_T$ parameterize a factorized Gaussian distribution over $\mathbf{y}_T$. The model is trained by maximizing the log-likelihood of the target set conditioned on the context set. In addition to the deterministic encoding of the context set, **Neural Process** (NP) [12] introduces a global latent variable $\mathbf{z}_G \in \mathbb{R}^{d_z}$ to capture task-level uncertainty. A separate neural network encodes $(\mathbf{x}_C, \mathbf{y}_C)$ by first aggregating the context set into a global summary $\mathbf{s}_C$, which is then used to parameterize a Gaussian distribution over $\mathbf{z}_G$. The decoder models the conditional distribution over $\mathbf{y}_T$ using $\mathbf{x}_T$, $\mathbf{r}_C$, and $\mathbf{z}_G$:

$$p(\mathbf{y}_T|\mathbf{x}_T, \mathbf{r}_C, \mathbf{s}_C) = \int p(\mathbf{y}_T|\mathbf{x}_T, \mathbf{r}_C, \mathbf{z}_G) \, p(\mathbf{z}_G|\mathbf{s}_C) \, d\mathbf{z}_G \tag{1}$$

The **Attentive Neural Process (AttnNP)** [25] extends NP by replacing the shared context representation $\mathbf{r}_C$ with a target-specific representation $\mathbf{r}_t$ computed using self-attention over the context and cross-attention with each target input $\mathbf{x}_t$. The decoder then conditions on $\mathbf{x}_t$, $\mathbf{r}_t$ and $\mathbf{z}_G$ to model the target distribution.

---

[1]For illustrative purposes, we used a special case where the bi-Lipschitz bounds were set to 1, resulting in an isometric mapping.

Figure 2: The encoder consists of global and local latent paths, each learning a distribution over its respective latent variables. The local path incorporates bi-Lipschitz regularization to approximately preserve input distances to model similarity relations among the data. The decoder then conditions on the sampled latents and the target embedding to predict the output.

While AttnNP uses attention to capture input-specific patterns, it does not explicitly preserve distance relationships among inputs. Its attention weights, computed over learned embeddings, may distort input relations [69, 55], hindering the capture of local dependencies crucial for similarity-based predictions. In contrast, while DNP also employs cross-attention to obtain target-specific latent variables, its key contribution is the construction of a distance-aware latent space via bi-Lipschitz regularization, from which attention weights are computed.

## 3 Distance-informed Neural Process

The Distance-informed Neural Process (DNP) is illustrated in Fig. 2. The encoder consists of both global and local latent paths. The global path models a distribution over the global latent variable $\mathbf{z}_G \in \mathbb{R}^{d_z}$, conditioned on the context data. To capture local dependencies, the distance-informed local path models a distribution over the target-specific latent variable $\mathbf{z}_t \in \mathbb{R}^{d_z}$, conditioned on the target input $\mathbf{x}_t$ and context data. The decoder then uses both the latent variables for the predictions.

### 3.1 Global latent path

The global latent variable $\mathbf{z}_G$ follows the standard formulation in NP and AttnNP, where it serves as a function-level representation that captures global uncertainty. The context set $(\mathbf{x}_C, \mathbf{y}_C)$ is first encoded into a summary representation $\mathbf{s}_C$ using mean aggregation over the encoded context pairs. This is further used to define the prior distribution as:

$$p_{\theta_G}(\mathbf{z}_G|\mathbf{x}_C, \mathbf{y}_C) = \mathcal{N}(\mu_{\theta_G}(\mathbf{s}_C), \Sigma_{\theta_G}(\mathbf{s}_C)), \tag{2}$$

where $\mu_{\theta_G}(\cdot)$ and $\Sigma_{\theta_G}(\cdot)$ are neural networks that parameterize the mean and diagonal covariance of the Gaussian distribution.

### 3.2 Local latent path

To derive the local latent variable, we encode the input points from both the context and target sets into low-dimensional latent representations using a neural network $h$. Each context point $\mathbf{x}_c \in \mathbf{x}_C$ is mapped to an embedding $\mathbf{u}_c = h(\mathbf{x}_c)$, and each target input $\mathbf{x}_t \in \mathbf{x}_T$ is mapped to $\mathbf{u}_t = h(\mathbf{x}_t)$,

where $\mathbf{u}_c, \mathbf{u}_t \in \mathbb{R}^{d_u}$. These representations are then used to model similarity relationships among the inputs via cross-attention.

Accurately capturing these similarity relations require that the latent embeddings preserve the geometric structure of the original input space. However, neural networks can distort this structure during encoding [55, 19], which compromises meaningful distance relationships in the latent space (see Fig. 1). Such distortions can lead to two key issues: (1) *over-sensitivity*, where small perturbations in the input cause disproportionately large changes in the latent space; and (2) *feature collapse*, where distinct inputs are mapped to similar embeddings, thereby reducing the discriminability in the learned representations.

**Definition 1** (Distortion bounded mapping). *A function $h : \mathcal{X} \to \mathcal{U}$ is said to be a distortion bounded mapping if there exist positive constants $L_1$ and $L_2$ such that, for all $\mathbf{x}_1, \mathbf{x}_2 \in \mathcal{X}$, the following bi-Lipschitz condition holds [45, 36]:*

$$L_1 \cdot d_{\mathcal{X}}(\mathbf{x}_1, \mathbf{x}_2) \leq d_{\mathcal{U}}(h(\mathbf{x}_1), h(\mathbf{x}_2)) \leq L_2 \cdot d_{\mathcal{X}}(\mathbf{x}_1, \mathbf{x}_2). \tag{3}$$

*Here, $d_{\mathcal{X}}(\mathbf{x}_1, \mathbf{x}_2)$ denotes the distance between $\mathbf{x}_1$ and $\mathbf{x}_2$ in the input space $\mathcal{X}$, and $d_{\mathcal{U}}(h(\mathbf{x}_1), h(\mathbf{x}_2))$ is the corresponding distance in the output (latent) space $\mathcal{U}$.*

The bi-Lipschitz mapping in Eq. 3, implemented by a neural network ensures that the latent space approximately preserves meaningful distances from the input manifold. The upper Lipschitz bound limits the sensitivity of the mapping, preventing small perturbations in the input from causing disproportionately large changes in the latent space. Conversely, the lower bound enforces sensitivity to meaningful variations by ensuring that distinct inputs remain distinguishable after mapping. Together, these constraints yield an approximately isometric transformation, promoting latent representations that remain faithful to the underlying geometric structure of the data.

Some well-known methods for enforcing bi-Lipschitz constraints in Euclidean space include two-sided gradient penalties [16, 56], orthogonal normalization [4], spectral normalization in architectures with residual connections [36, 39], and reversible models [21, 2]. While these techniques are used in the context of GANs [39, 16] and deep kernel learning [55], applying bi-Lipschitz regularization to NPs is, to our knowledge, novel. Among the existing approaches, spectral normalization offers the best empirical trade-off between stability and computational cost [55]. For architectures with residual connections, bounding the largest singular values of the weight matrices via spectral normalization [39, 68], combined with Lipschitz-continuous activation functions is sufficient to achieve the bi-Lipschitz property of the network [2]. However, for more general architectures without residual links, we extend this approach by explicitly constraining both the smallest and largest singular values of each weight matrix to directly control the network's contraction and expansion behavior across layers.

**Weight Regularization.** A neural network typically consists of a series of $L$ layers, where each layer applies a linear transformation followed by a non-linear activation function $a$ as $g_l(\mathbf{x}) = a(\mathbf{W}_l\mathbf{x}+\mathbf{b}_l)$. To encourage bi-Lipschitz behavior, both the linear transformation and the activation function must together form a bi-Lipschitz mapping. For the linear part, this is achieved by bounding the maximum and minimum singular values of the weight matrices $\mathbf{W}_l$. To ensure overall bi-Lipschitz continuity, this bounding is combined with bi-Lipschitz activation functions such as Leaky ReLU [38], Softplus, ELU [6], PReLU [18], etc which preserve both upper and lower Lipschitz bounds. Note that residual architectures using spectral normalization [2] have similar requirements, as they rely on Lipschitz-continuous activations to ensure the bi-Lipschitz property. Let $\sigma^l_{\min}$ and $\sigma^l_{\max}$ denote the smallest and largest singular values of $\mathbf{W}_l$ respectively. These values control how much the input can be contracted or stretched by the linear transformation. Accordingly, we define the bi-Lipschitz regularization loss:

$$\mathcal{L}_{\text{bi-Lip}} = \sum_{l=1}^{L} \max(0, \lambda_1 - \sigma^l_{\min})^2 + \max(0, \sigma^l_{\max} - \lambda_2)^2, \tag{4}$$

where $0 < \lambda_1 \leq \lambda_2$ are hyperparameters that control the lower and upper bounds of $\sigma^l_{\min}$ and $\sigma^l_{\max}$, respectively. These hyperparameters provide flexibility in managing the effective Lipschitz constants when combined with regularization techniques (e.g., Dropout, Batch Normalization). The overall Lipschitz constants of the function $h = g_L \circ g_{L-1} \circ \cdots \circ g_1$ are governed by the product of the per-layer singular value bounds.

Note that computing the exact singular value decomposition (SVD) of each weight matrix can be computationally expensive, particularly for deep networks. For a matrix $\mathbf{W} \in \mathbb{R}^{p \times q}$, the computa-

tional cost of SVD is $\mathcal{O}(\min(pq^2, p^2q))$, which can be prohibitive for large $p$ and $q$. To obtain a fast approximation of $\sigma^l_{\max}$ and $\sigma^l_{\min}$, we apply the Locally Optimal Block Preconditioned Conjugate Gradient (LOBPCG) method [29, 51], an iterative algorithm originally designed to compute a few extremal eigenvalues and corresponding eigenvectors of large symmetric matrices. In our case, LOBPCG is used to approximate the largest and smallest eigenvalues of the symmetric positive semi-definite matrix $\mathbf{W}_l \mathbf{W}_l^\top$ or $\mathbf{W}_l^\top \mathbf{W}_l$, depending on which of the two is smaller in dimension. For $T$ iterations, the total computational cost per layer is $\mathcal{O}(Tpq)$, where $T \approx 5 - 10$. The approximate values of $\sigma^l_{\max}$ and $\sigma^l_{\min}$ are then obtained as the square roots of these eigenvalues respectively.

**Similarity relations.** Once the latent space embeddings are obtained, we use cross-attention [57] to model the relationships between the context and target points. The latent embeddings of the context data $\mathbf{u}_c$ serve as the keys, while the embedding of a target point $\mathbf{u}_t$ acts as the query. The cross-attention weights are computed using Laplace attention as follows:

$$\alpha^c_t = \frac{\exp\left(-\frac{||\mathbf{u}_t - \mathbf{u}_c||}{\sqrt{d_u}}\right)}{\sum_{c' \in C} \exp\left(-\frac{||\mathbf{u}_t - \mathbf{u}_{c'}||}{\sqrt{d_u}}\right)}. \tag{5}$$

The attention mechanism assigns a relevance score to each context point based on its similarity to the target point. Since the embeddings are constructed to preserve the geometric structure of the input space, the resulting attention weights more accurately reflect the true proximity between points.

**Local latent variables.** The target-specific prior distribution for the local latent variable $\mathbf{z}_t$ is a Gaussian and follows a similar construction to [37]:

$$p_{\theta_L}(\mathbf{z}_t | \mathbf{x}_t, \mathbf{x}_C, \mathbf{y}_C) = \mathcal{N}\left(\sum_{c \in C} \alpha^c_t \mu_{\theta_L}(\mathbf{x}_c, \mathbf{y}_c), \sum_{c \in C} \exp(\alpha^c_t \Sigma_{\theta_L}(\mathbf{x}_c, \mathbf{y}_c))\right). \tag{6}$$

$\mu_{\theta_L}(\cdot, \cdot)$ and $\Sigma_{\theta_L}(\cdot, \cdot)$ are neural networks used to parameterize the Gaussian distribution. The attention weight $\alpha^c_t$ control how much each context point contributes to the local latent distribution. When a target point lies in an OOD region and far from the context data, the attention weights tend toward zero. As a result, the local prior distribution over $\mathbf{z}_t$ approaches a standard normal, functioning as a non-informative prior.

### 3.3 Generative Model

Once the global and local latent variables are defined, we construct the decoder (likelihood model) that generates outputs $\mathbf{y}_{1:N}$ from inputs $\mathbf{x}_{1:N}$. This decoder network is conditioned on both the global latent $\mathbf{z}_G$ and the local latent $\mathbf{z}_{1:N}$ corresponding to $\mathbf{x}_{1:N}$, and is given by the conditional distribution $p_\theta(\mathbf{y}_{1:N} | \mathbf{z}_G, \mathbf{z}_{1:N}, \mathbf{x}_{1:N})$. The complete generative process is given by:

$$p_{\mathbf{x}_{1:N}}(\mathbf{y}_{1:N}) = \iint \prod_{i=1}^N p_\theta(\mathbf{y}_i | \mathbf{z}_G, \mathbf{z}_i, \mathbf{x}_i)\, p_{\theta_L}(\mathbf{z}_i | \mathbf{x}_i)\, p_{\theta_G}(\mathbf{z}_G)\, d\mathbf{z}_{1:N}\, d\mathbf{z}_G \tag{7}$$

Note that we choose factorized Gaussian priors for $\mathbf{z}_G$ and each $\mathbf{z}_i$ for simplicity. This choice does not limit the generality of our approach: any permutation-invariant distribution over the global and local latents could be substituted in its place.

**Proposition 1.** *Eq. 7 defines an exchangeable stochastic process by satisfying both exchangeability and marginal consistency, as required by the Kolmogorov Extension Theorem [41].*

The proof is provided in Appendix C. Intuitively, exchangeability holds because the likelihood and local priors factorize across data points, so permuting the input-output pairs does not change the joint distribution. Marginal consistency holds because integrating out a subset of outputs and their corresponding latents simply removes those terms from the product, yielding a valid joint distribution over the remaining variables.

### 3.4 Training and Inference

Having defined the generative model, we aim to fit the model parameters to infer the latent variables, and to make predictions for new inputs $\mathbf{x}^*$. However, exact inference using Eq. 7 is intractable.

To address this, we employ variational inference, which involves maximizing the Evidence Lower Bound (ELBO) on the log marginal likelihood [28]. We introduce the variational distributions $q_{\phi_G}(\mathbf{z}_G|\mathbf{x}_T, \mathbf{y}_T)$ and $q_{\phi_L}(\mathbf{z}_t|\mathbf{x}_t, \mathbf{y}_t)$ to approximate the true posterior distributions over the global and local latent variables, respectively. The global variational posterior is modeled as a Gaussian:

$$q_{\phi_G}(\mathbf{z}_G|\mathbf{x}_T, \mathbf{y}_T) = \mathcal{N}(\mu_{\phi_G}(\mathbf{s}_T), \Sigma_{\phi_G}(\mathbf{s}_T)), \tag{8}$$

where $\mathbf{s}_T$ is the mean-aggregated summary of the target set, obtained in the same manner as $\mathbf{s}_C$. The local variational posterior $q_{\phi_L}$ is also modeled as a Gaussian with diagonal covariance, parameterized by neural networks:

$$q_{\phi_L}(\mathbf{z}_t|\mathbf{x}_t, \mathbf{y}_t, \mathbf{x}_C, \mathbf{y}_C) = \mathcal{N}(\mu_{\phi_L}(\mathbf{x}_t, \mathbf{y}_t, \mathbf{x}_C, \mathbf{y}_C), \Sigma_{\phi_L}(\mathbf{x}_t, \mathbf{y}_t, \mathbf{x}_C, \mathbf{y}_C))). \tag{9}$$

The resulting ELBO, derived in Appendix D is:

$$\begin{aligned}
\mathcal{L}_{\text{ELBO}} = \mathbb{E}_{q_{\phi_G} q_{\phi_L}}[\log p_\theta(\mathbf{y}_t|\mathbf{x}_t, \mathbf{z}_t, \mathbf{z}_G)] &- D_{KL}(q_{\phi_G}(\mathbf{z}_G|\mathbf{x}_T, \mathbf{y}_T)||p_{\theta_G}(\mathbf{z}_G|\mathbf{x}_C, \mathbf{y}_C)) \\
&- D_{KL}(q_{\phi_L}(\mathbf{z}_t|\mathbf{x}_t, \mathbf{y}_t, \mathbf{x}_C, \mathbf{y}_C)||p_{\theta_L}(\mathbf{z}_t|\mathbf{x}_t, \mathbf{x}_C, \mathbf{y}_C))
\end{aligned} \tag{10}$$

The final training objective of DNP is the sum of the ELBO loss and the bi-Lipschitz regularization loss:

$$\mathcal{L} = \mathcal{L}_{\text{ELBO}} + \beta \mathcal{L}_{\text{bi-Lip}}, \tag{11}$$

where $\beta$ is a hyperparameter that control the trade-off between data likelihood and weight regularization. For inference on unseen data points $\mathbf{x}^*$, we use the posterior predictive distribution:

$$p(\mathbf{y}^*|\mathbf{x}^*, \mathbf{x}_C, \mathbf{y}_C) = \iint p_\theta(\mathbf{y}^*|\mathbf{z}_G, \mathbf{z}^*, \mathbf{x}^*) p_{\theta_L}(\mathbf{z}^*|\mathbf{x}^*, \mathbf{x}_C, \mathbf{y}_C) p_{\theta_G}(\mathbf{z}_G|\mathbf{x}_C, \mathbf{y}_C) d\mathbf{z}^* d\mathbf{z}_G. \tag{12}$$

Note that at test time, the learned prior networks $p_{\theta_G}$ and $p_{\theta_L}$ are used to generate the latent variables and for the prediction. The algorithms and computational complexity for training and inference is provided in Appendix. A.1 and A.2.

## 4 Related Works

Stochastic Neural Networks (SNNs) have been widely explored for capturing uncertainty in neural networks. Bayesian Neural Networks (BNNs) [3] introduce a prior distribution over weights and use Bayesian inference to estimate posterior distributions, and estimate uncertainty in deep learning models. However, BNNs are computationally expensive, leading to the development of approximation techniques such as variational inference [15, 47], Monte Carlo dropout [10] and deep ensembles [32]. While BNNs approximate posterior distributions over weights, an alternative Bayesian approach is provided by Gaussian Processes (GPs)[66], which define distributions over functions directly. Unlike BNNs, GPs offer a non-parametric framework for capturing uncertainty, making them particularly appealing for small-data regimes. However, the cubic computational complexity of standard GPs limits their scalability, and they are less flexible in high-dimensional data, motivating various approximations such as sparse GPs [54, 49] and deep kernel learning [67].

Neural Processes (NPs) [11, 12, 25] have emerged as a powerful class of models that combine the flexibility of deep learning with the uncertainty quantification of Bayesian inference. In addition to the NPs discussed in Sec. 2, several other variants have been introduced to model different inductive biases. Bootstrapping Neural Process (BNP) [34] improves uncertainty estimation in NPs by incorporating a bootstrapping procedure to iteratively improve latent variable estimates. Convolutional CNP (ConvCNP) [14] and ConvNP [8] introduce translation equivariance by leveraging convolutional neural networks, which improves generalization in tasks involving spatial dependencies such as time-series. Functional NP [37] leverages graph structures to model data dependencies, thereby obtaining more localized latent representations. In contrast, DSVNP [63] introduces a hierarchical structure of global and local latent variables to capture uncertainty at multiple levels of granularity. Transformer Neural Processes [40] replace the standard NP encoder–decoder with a transformer, improving expressiveness while ensuring context invariance and target equivariance. [64] extend NPs with

multiple expert latent variables to better capture multimodal distributions. [52] reformulate context aggregation as probabilistic graphical inference, yielding mixture and robust Bayesian aggregation variants that improve OOD robustness. [62] address NP underfitting via a surrogate objective in an expectation–maximization framework, leading to more accurate distribution learning. [13] propose a decoupled encoder trained with a contrastive objective, enhancing transferability and robustness across tasks. While these works focus on expressiveness and inference, DNP takes a complementary path by enforcing geometry-aware regularization through a bi-Lipschitz-constrained latent path. This preserves input structure and improves uncertainty estimation and OOD generalization. Following [63], a summary of selected NP variants is provided in Table 1.

Table 1: Comparison of different Neural Process (NP) variants.

| NP Family | Recognition Model | Generative Model | Prior Distribution | Latent Variable |
|---|---|---|---|---|
| CNP | $\mathbf{z}_C = f(\mathbf{x}_C, \mathbf{y}_C)$ | $p(\mathbf{y}_T \vert \mathbf{z}_C, \mathbf{x}_T)$ | - | Global (Deterministic) |
| NP | $q(\mathbf{z}_G \vert \mathbf{x}_T, \mathbf{y}_T)$ | $p(\mathbf{y}_T \vert \mathbf{z}_G, \mathbf{x}_T)$ | $p(\mathbf{z}_G \vert \mathbf{x}_C, \mathbf{y}_C)$ | Global (Stochastic) |
| ConvCNP | $\mathbf{z}_i = f(\mathbf{x}_C, \mathbf{y}_C, \mathbf{x}_i)$ | $p(\mathbf{y}_i \vert \mathbf{z}_i)$ | - | Local (Deterministic) |
| ConvNP | $\mathbf{z}_i \sim Enc_\phi(\mathbf{x}_C, \mathbf{y}_C, \mathbf{x}_i)$ | $p(\mathbf{y}_i \vert \mathbf{z}_i)$ | Implicit via encoder | Local (Stochastic) |
| AttnNP | $q(\mathbf{z}_G \vert \mathbf{x}_T, \mathbf{y}_T)$ $\mathbf{z}_i = f(\mathbf{x}_C, \mathbf{y}_C, \mathbf{x}_i)$ | $p(\mathbf{y}_i \vert \mathbf{z}_G, \mathbf{z}_i, \mathbf{x}_i)$ | $p(\mathbf{z}_G \vert \mathbf{x}_C, \mathbf{y}_C)$ | Global (Stochastic) + Local (Deterministic) |
| DSVNP | $q(\mathbf{z}_G \vert \mathbf{x}_T, \mathbf{y}_T)$, $q(\mathbf{z}_i \vert \mathbf{z}_G, \mathbf{x}_i, \mathbf{y}_i)$ | $p(\mathbf{y}_i \vert \mathbf{z}_G, \mathbf{z}_i, \mathbf{x}_i)$ | $p(\mathbf{z}_G \vert \mathbf{x}_C, \mathbf{y}_C)$, $p(\mathbf{z}_i \vert \mathbf{z}_G, \mathbf{x}_i)$ | Global (Stochastic) + Local (Stochastic) |
| DNP (Ours) | $q(\mathbf{z}_G \vert \mathbf{x}_T, \mathbf{y}_T)$, $q(\mathbf{z}_i \vert \mathbf{x}_C, \mathbf{y}_C, \mathbf{x}_i)$ | $p(\mathbf{y}_i \vert \mathbf{z}_G, \mathbf{z}_i, \mathbf{x}_i)$ | $p(\mathbf{z}_G \vert \mathbf{x}_C, \mathbf{y}_C)$, $p(\mathbf{z}_i \vert \mathbf{x}_C, \mathbf{y}_C, \mathbf{x}_i)$ | Global (Stochastic) + Local (Stochastic) |

# 5    Experiments

We evaluate DNP on both regression and classification tasks. For regression, we consider 1D synthetic examples, as well as multi-dimensional tasks on several real-world datasets. Performance is assessed in terms of predictive quality and uncertainty calibration [31]. For classification, we assess the model on CIFAR-10 and CIFAR-100 [30], evaluating classification accuracy, calibration performance, and OOD detection capabilities. The regression networks use a 3-layer MLP, while a VGG-16 architecture [30] is used for feature extraction in classification. The data generation process for the experiments, model architectures used and the hyperparameters are provided in Appendix A.

## 5.1    1D Synthetic Regression

In this experiment, we evaluate the methods on 1D functions generated from GP priors with three different kernels: (i) the Radial Basis Function (RBF) kernel, (ii) the Matérn-5/2 kernel, and (iii) the Periodic kernel. At each training iteration, kernel hyperparameters are randomly sampled to introduce functional diversity. During testing, the methods are evaluated on 2000 function realizations, using the log-likelihood (LL) and Expected Calibration Error (ECE) [31] of the target points.

Table 2 shows the results for evaluation on the target set, excluding the context. Across all three kernel types, DNP achieves a superior log-likelihood and a lower ECE scores in majority of the cases compared to the other baselines. Evaluation on the context set, inference latency and noisy data is provided in Appendix B.1. Figure 3 shows predictions from AttnNP and DNP on GP-sampled functions, evaluated both in-distribution ($[-2, 2]$) and in OOD regions. While AttnNP often underestimates uncertainty, DNP is generally better calibrated. DNP can occasionally be overconfident due to the approximate nature of the bi-Lipschitz constraint. However, DNP achieves more reliable predictions, accurately modeling the target function in-distribution while expressing higher uncertainty in OOD regions.

## 5.2    Synthetic to Real-World Regression

Following [14], we consider the Lotka–Volterra model [65], which captures predator–prey population dynamics. Models are trained on simulated data generated from the Lotka–Volterra equations and

Table 2: Performance comparison for 1D synthetic regression. Results are averaged over 10 runs.

| Method | RBF | | Matérn-$\frac{5}{2}$ | | Periodic | |
|---|---|---|---|---|---|---|
| | LL ($\uparrow$) | ECE ($\downarrow$) | LL ($\uparrow$) | ECE ($\downarrow$) | LL ($\uparrow$) | ECE ($\downarrow$) |
| CNP [11] | 0.490±0.006 | 0.122±0.061 | 0.419±0.004 | 0.088±0.047 | 0.239±0.001 | 0.120±0.029 |
| NP [12] | 0.480±0.009 | 0.102±0.082 | 0.318±0.008 | 0.111±0.053 | 0.228±0.002 | 0.145±0.047 |
| ConvCNP [14] | 0.496±0.007 | 0.118±0.054 | 0.329±0.001 | 0.098±0.032 | 0.335±0.002 | 0.106±0.032 |
| ConvNP [8] | 0.609±0.005 | 0.190±0.074 | 0.313±0.002 | 0.137±0.095 | 0.431±0.003 | 0.117±0.011 |
| AttnNP [25] | 0.578±0.003 | 0.103±0.026 | 0.499±0.009 | **0.078±0.014** | 0.457±0.001 | 0.098±0.038 |
| DSVNP [63] | 0.582±0.002 | 0.101±0.032 | 0.512±0.006 | 0.101±0.012 | 0.412±0.002 | 0.092±0.042 |
| NP-mBA [52] | 0.521±0.005 | 0.108±0.012 | 0.324±0.007 | 0.091±0.021 | 0.375±0.005 | 0.091±0.010 |
| Ours | **0.696±0.003** | **0.093±0.054** | **0.620±0.002** | 0.083±0.010 | **0.635±0.001** | **0.074±0.022** |

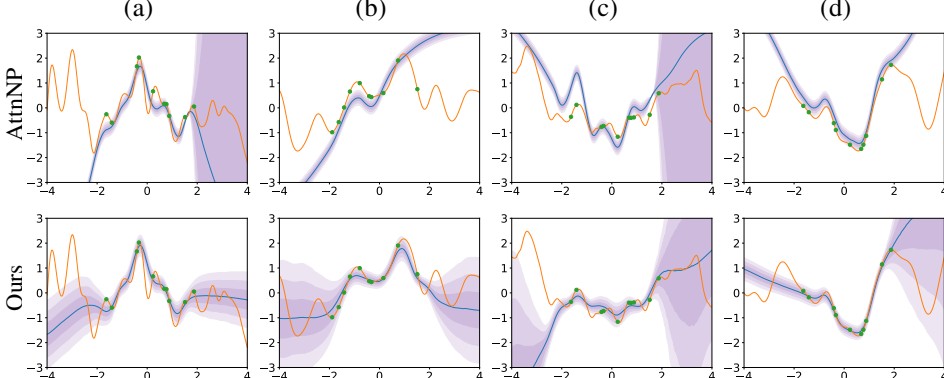

Figure 3: Predictive distributions for sample realizations from the GP kernel. The model is trained on inputs from the region [2,2]. The orange curve represents the ground truth, and the green points are the observations. The blue curve shows the model's mean prediction. Shaded regions indicate the ±3 standard deviation confidence interval.

evaluated on real-world data from Hudson's Bay hare–lynx data [35], presenting a distribution mismatch scenario. For each training batch, the number of context points is sampled uniformly from the interval [3, 50], while the number of target points is fixed at 50. During testing, 30 context points are randomly selected from the training set. In addition to NPs, we also include an exact GP with RBF kernel as a baseline. Table 3 reports log-likelihood scores on both simulated and real data. Results indicate that DNP achieves a better performance over the other baselines. Interestingly, the exact GP performs worse than the NP variants, which can be attributed to the limited context points. In contrast, NP-based methods are able to more efficiently incorporate context information, resulting in higher-quality predictions.

## 5.3 Multi-output Real-World Regression

To assess the performance of DNP on more complex and practical scenarios, we follow [63] and evaluate the models on the SARCOS [61], Water Quality (WQ) [7] and the SCM20D [50] datasets. The SARCOS dataset contains 48,933 samples, each with 21 input features and 7 output variables. The Water Quality (WQ) dataset includes 1,060 samples with 16 input features and 14 output variables. The SCM20D dataset comprises 8,966 samples, with 61 inputs and 16 outputs. Performance is measured using Mean Squared Error (MSE) and ECE scores. The results are provided in Table 4, which shows that DNP achieves superior predictive performance in most cases and provides consistently improved uncertainty calibration across all three datasets.

Table 3: Log-likelihood values for the predator-prey experiment. Results are averaged over 10 runs.

| Method | GP(Exact) | CNP | NP | ConvNP | AttnNP | DSVNP | Ours |
|---|---|---|---|---|---|---|---|
| Simulated | 0.952±0.024 | 1.293±0.034 | 1.382±0.036 | 1.608±0.043 | 1.590±0.019 | 1.507±0.027 | **1.712±0.029** |
| Real | -9.632±0.018 | -6.435±0.024 | -5.287±0.019 | -5.192±0.013 | -5.003±0.022 | -4.897±0.031 | **-2.967±0.027** |

Table 4: Prediction performance and uncertainty calibration on multi-output real-world regression datasets, averaged over 10 runs.

| Method | Metrics | GP (exact) | CNP [11] | NP [12] | AttnNP [25] | DSVNP [63] | Ours |
|--------|---------|-----------|----------|---------|-------------|------------|------|
| SARCOS | MSE ($\downarrow$) | 0.998±0.028 | 0.940±0.021 | 1.093±0.029 | 1.004±0.032 | 0.938±0.018 | **0.886±0.012** |
|        | ECE ($\downarrow$) | 0.102±0.012 | 0.113±0.013 | 0.105±0.018 | 0.108±0.015 | 0.102±0.012 | **0.083±0.011** |
| WQ     | MSE ($\downarrow$) | 0.721±0.008 | 0.739±0.010 | 0.724±0.005 | **0.692±0.005** | 0.723±0.003 | 0.715±0.006 |
|        | ECE ($\downarrow$) | 0.089±0.007 | 0.087±0.009 | 0.100±0.013 | 0.126±0.008 | 0.108±0.012 | **0.084±0.009** |
| SCM20D | MSE ($\downarrow$) | 0.932±0.008 | 0.956±0.004 | 0.962±0.005 | 0.936±0.003 | 0.912±0.004 | **0.887±0.004** |
|        | ECE ($\downarrow$) | 0.109±0.008 | 0.124±0.005 | 0.131±0.003 | 0.122±0.007 | 0.119±0.002 | **0.088±0.003** |

## 5.4 Image Classification

We evaluate the performance of DNP on image classification using the CIFAR-10 and CIFAR-100 datasets. The models are assessed in terms of classification accuracy, ECE, and the predictive entropy. Additionally, we evaluate OOD detection performance using the Area Under the Precision-Recall curve (AUPR) between the ID and the OOD probabilities. For classification using CIFAR10, we use the SVHN [46], CIFAR100 and the TinyImageNet [33] as the OOD datasets. For CIFAR100, the OOD datasets are SVHN, CIFAR10 and TinyImageNet. The VGG16 model [48] is used as the feature extractor for all the methods. In addition to the NP-based baselines, we also provide the results for a VGG-16 classifier trained using cross-entropy loss, referred to as "Deterministic".

Table 5: Image classification using CIFAR10 (ID) vs SVHN/CIFAR100/TinyImageNet (OOD). Results are averaged over 10 runs.

| Dataset | Metrics | Deterministic | CNP | NP | AttnNP | DSVNP | Ours |
|---------|---------|---------------|-----|----|--------|-------|------|
| CIFAR10 | Accuracy ($\uparrow$) | **92.08±0.01** | 89.72±0.02 | 90.86±0.05 | 90.37±0.01 | 90.41±0.03 | 91.97±0.01 |
|         | ECE ($\downarrow$) | 0.032±0.002 | 0.034±0.001 | 0.025±0.001 | 0.020±0.000 | 0.018±0.001 | **0.011±0.002** |
|         | Entropy ($\downarrow$) | **0.122±0.001** | 0.196±0.011 | 0.179±0.000 | 0.237±0.000 | 0.201±0.001 | 0.209±0.000 |
|         | Latency (ms) ($\downarrow$) | **3.526±0.104** | 3.679±0.120 | 6.011±0.113 | 8.207±0.135 | 8.432±0.109 | 6.716±0.122 |
| SVHN    | AUPR ($\uparrow$) | 76.34±0.01 | 89.68±0.02 | 85.66±0.01 | 93.69±0.02 | 93.42±0.01 | **97.05±0.01** |
|         | Entropy ($\uparrow$) | 0.134±0.006 | 1.022±0.006 | 0.951±0.002 | 1.419±0.001 | 1.816±0.002 | **2.279±0.001** |
| CIFAR100 | AUPR ($\uparrow$) | 80.96±0.01 | 87.23±0.01 | 87.33±0.01 | 87.11±0.01 | 87.14±0.01 | **90.58±0.01** |
|         | Entropy ($\uparrow$) | 0.173±0.005 | 0.790±0.004 | 0.735±0.001 | 0.891±0.001 | 0.912±0.001 | **1.167±0.001** |
| TinyImageNet | AUPR ($\uparrow$) | 74.72±0.01 | 85.74±0.01 | 86.48±0.01 | 86.34±0.01 | 87.23±0.01 | **90.04±0.01** |
|         | Entropy ($\uparrow$) | 0.168±0.005 | 0.808±0.003 | 0.761±0.002 | 0.900±0.000 | 0.912±0.001 | **1.197±0.001** |

Table 6: Image classification using CIFAR100 (ID) vs SVHN/CIFAR10/TinyImageNet (OOD). Results are averaged over 10 runs.

| Dataset | Metrics | Deterministic | CNP | NP | AttnNP | DSVNP | Ours |
|---------|---------|---------------|-----|----|--------|-------|------|
| CIFAR100 | Accuracy ($\uparrow$) | **70.02±0.01** | 68.74±0.05 | 69.26±0.01 | 69.30±0.07 | 69.28±0.04 | 69.30±0.04 |
|          | ECE ($\downarrow$) | 0.136±0.002 | 0.103±0.001 | 0.119±0.001 | 0.118±0.009 | 0.104±0.002 | **0.082±0.003** |
|          | Entropy ($\downarrow$) | **0.258±0.001** | 0.834±0.001 | 0.805±0.000 | 0.663±0.011 | 0.672±0.007 | 0.677±0.012 |
|          | Latency (ms) ($\downarrow$) | **3.662±0.110** | 3.825±0.116 | 6.228±0.117 | 8.424±0.117 | 8.637±0.125 | 6.876±0.118 |
| SVHN     | AUPR ($\uparrow$) | 75.32±0.02 | 84.62±0.01 | 85.34±0.01 | 87.01±0.01 | 86.28±0.01 | **89.92±0.01** |
|          | Entropy ($\uparrow$) | 0.298±0.003 | 1.424±0.003 | 0.937±0.003 | 1.221±0.002 | 1.292±0.001 | **3.494±0.002** |
| CIFAR10  | AUPR ($\uparrow$) | 70.64±0.02 | 73.49±0.01 | 76.32±0.01 | 69.57±0.01 | 76.33±0.01 | **80.56±0.01** |
|          | Entropy ($\uparrow$) | 0.282±0.003 | 1.515±0.002 | 1.573±0.003 | 1.225±0.003 | 1.232±0.004 | **2.548±0.003** |
| TinyImageNet | AUPR ($\uparrow$) | 72.73±0.01 | 74.84±0.01 | 75.46±0.01 | 73.41±0.01 | 76.82±0.01 | **80.09±0.01** |
|          | Entropy ($\uparrow$) | 0.306±0.003 | 1.431±0.002 | 1.534±0.003 | 1.286±0.004 | 1.291±0.003 | **2.441±0.003** |

Table 5 shows the results for classification uisng CIFAR10. From the results, DNP achieves the least ECE score, and closely follows Deterministic in terms of accuracy. DNP incurs higher inference latency per instance compared to CNP and vanilla NP. However, when compared to AttnNP and DSVNP, DNP achieves a faster inference by avoiding self-attention over the context set, instead relying on a bi-Lipschitz regularized latent space to preserve the relational structure. Interestingly, the in-distribution entropy of DNP is slightly higher than that of the vanilla NP. This could be attributed to DNP's more expressive uncertainty modeling, where the combination of global and local latent variables introduces additional variance to capture the predictive uncertainty. Consequently, DNP also shows increased entropy on the OOD datasets, allowing it to more effectively distinguish ID from OOD samples. This leads to improved performance in OOD detection, as reflected in its higher AUPR

scores compared to other methods. A similar trend is observed in the classification performance on CIFAR-100, as shown in Table 6. DNP achieves a lower ECE, while also outperforming all baselines in OOD detection.

To evaluate the contribution of different regularization methods aimed at controlling the Lipschitz constant of the network, we compare DNP variants using the 2-sided gradient penalty [16], orthogonal regularization [4], and spectral regularization [68] for CIFAR10 (ID) vs CIFAR100 (OOD) classification. The results in Table 7 show that while all regularization methods improve ECE and AUPR scores compared to the no-

Table 7: Comparison of regularization methods in DNP on accuracy, calibration (ECE), and OOD detection (AUPR). Results are averaged over 10 runs.

| Regularization | Accuracy ($\uparrow$) | ECE ($\downarrow$) | AUPR ($\uparrow$) |
|---|---|---|---|
| DNP w/o reg. | **92.08**±**0.01** | 0.042±0.001 | 87.85±0.01 |
| 2-sided Gradient Penalty | 90.90±0.02 | 0.032±0.001 | 88.92±0.01 |
| Orthogonal reg. | 90.96±0.01 | 0.034±0.000 | 88.64±0.01 |
| Spectral Norm reg. | 91.42±0.01 | 0.015±0.001 | 89.73±0.01 |
| Bi-Lipschitz reg. (ours) | 91.97±0.01 | **0.011**±**0.002** | **90.58**±**0.01** |

regularization baseline, bi-Lipschitz weight regularization consistently outperforms the others. Notably, both spectral and bi-Lipschitz regularization yield a substantial reduction in ECE, with the latter achieving the lowest ECE and highest AUPR scores. In contrast, methods like orthogonal regularization are more restrictive, as they impose stronger constraints on the model's weight space, which may limit representational flexibility. Additional ablations on the influence of the number of context points, dimensions of $d_u$ and $d_z$, $\lambda_1, \lambda_2$ values for the bi-Lipschitz regularization, trade off parameter $\beta$ and comparison with dot-product attention are provided in Appendix B.2.

## 6 Limitations and Future Work

The current bi-Lipschitz constraint is applied in a Euclidean space, which may not fully capture complex semantic relationships. Extending this constraint to incorporate more flexible distance measures or task-specific similarity metrics [69] could improve generalization across diverse domains. Moreover, the use of cross-attention introduces additional computational overhead, which may limit scalability to very large datasets. Exploring sparse attention mechanisms [22, 53] offers a promising path to reduce this cost while maintaining expressiveness.

## 7 Discussion and Conclusion

We introduced the Distance-informed Neural Process (DNP), a novel variant within the Neural Process family that integrates both global and local latent representations. DNP leverages a distance-aware local latent path, enforced via bi-Lipschitz regularization to approximately preserve distance relationships in the input space. This leads to a more faithful modeling of similarity relationships and improves both predictive accuracy and uncertainty calibration across regression and classification tasks. The broader implications of this work lie in its potential applications in safety-critical domains, where calibrated uncertainty estimates and generalization to unseen conditions are essential. Like most neural models, DNP can be sensitive to the quality and bias of its training data and dataset shifts. We encourage practitioners to validate the model carefully to ensure safe and reliable deployment.

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

# A   Implementation Details

## A.1   Algorithm

The training and inference procedures for DNP is outlined in Algorithms 1 and 2 respectively. The model combines a global latent variable that captures task-level patterns and a distance-aware local latent variable that models input similarity. During training, both the global and local paths are optimized jointly using a variational objective and a bi-Lipschitz regularization term. At test time, the model predicts target outputs by conditioning on both the global and local priors inferred from the context set.

---

**Algorithm 1** Training DNP

---

**Input** : Mini-batch $(\mathbf{x}, \mathbf{y})_{i=1}^{B}$ from dataset $\mathcal{D}$, trade-off parameter $\beta$, minimum and maximum number of context data $N_{\min}, N_{\max}$.
**Output** : Updated parameters $\theta, \theta_G, \theta_L, \phi_G, \phi_L$
**foreach** *mini-batch* **do**

    Randomly draw the number of context points $N_c \sim \mathcal{U}(N_{\min}, N_{\max})$;
    Sample a context set $(\mathbf{x}_C, \mathbf{y}_C) \subset (\mathbf{x}, \mathbf{y})$ of size $N_c$, define the target set as $(\mathbf{x}_T, \mathbf{y}_T) = (\mathbf{x}, \mathbf{y})$;
    Compute global summary: $s_C = \text{mean}(\text{NN}(\mathbf{x}_c, \mathbf{y}_c))$;
    Compute prior $p_{\theta_G}(\mathbf{z}_G | \mathbf{x}_C, \mathbf{y}_C)$;
    Encode inputs: $\mathbf{u}_c = h(\mathbf{x}_c)$, $\mathbf{u}_t = h(\mathbf{x}_t)$;
    Compute attention weights $\alpha_t^c$ using dot-product attention;
    Compute local prior $p_{\theta_L}(\mathbf{z}_t | \mathbf{x}_t, \mathbf{x}_C, \mathbf{y}_C)$;
    Compute variational posteriors: $q_{\phi_G}(\mathbf{z}_G | \mathbf{x}_T, \mathbf{y}_T)$ and $q_{\phi_L}(\mathbf{z}_t | \mathbf{x}_t, \mathbf{y}_t)$;
    Sample $\mathbf{z}_G \sim q_{\phi_G}$ and $\mathbf{z}_t \sim q_{\phi_L}$;
    Compute likelihood $p_\theta(\mathbf{y}_t | \mathbf{x}_t, \mathbf{z}_t, \mathbf{z}_G)$;
    Compute $\mathcal{L}_{\text{ELBO}}$ (Eq. 10);
    Compute $\mathcal{L}_{\text{bi-Lip}}$ (Eq. 4);
    Total loss: $\mathcal{L} = \mathcal{L}_{\text{ELBO}} + \beta \mathcal{L}_{\text{bi-Lip}}$;
    Update model parameters via gradient descent;
**end**

---

**Algorithm 2** Inference with DNP

---

**Input** : Context set $(\mathbf{x}_C, \mathbf{y}_C)$, test input $\mathbf{x}^*$, trained parameters
**Output** : Predictive distribution $p(\mathbf{y}^* | \mathbf{x}^*, \mathbf{x}_C, \mathbf{y}_C)$
Encode: $\mathbf{u}_c = h(\mathbf{x}_c)$, $\mathbf{u}^* = h(\mathbf{x}^*)$;
Compute attention weights $\alpha_c^*$ using dot-product attention;
Sample $\mathbf{z}_G \sim p_{\theta_G}(\mathbf{z}_G | \mathbf{x}_C, \mathbf{y}_C)$;
Sample $\mathbf{z}^* \sim p_{\theta_L}(\mathbf{z}^* | \mathbf{x}^*, \mathbf{x}_C, \mathbf{y}_C)$;
Return $p_\theta(\mathbf{y}^* | \mathbf{x}^*, \mathbf{z}^*, \mathbf{z}_G)$;

---

## A.2   Computational Complexity

CNPs and NPs aggregate the context set using a permutation-invariant operation (e.g., mean pooling), resulting in an $\mathcal{O}(M)$ operation. Each of the $N$ target points is then processed independently, yielding a total prediction-time complexity of $\mathcal{O}(N)$. AttnNPs and DSVNP uses self-attention over the $M$ context points, which incurs a cost of $\mathcal{O}(M^2)$. Additionally, they compute cross-attention between each of the $N$ targets and the $M$ context points, resulting in a further $\mathcal{O}(NM)$ cost. Thus, the overall complexity during prediction becomes $\mathcal{O}(M^2 + NM)$. Our proposed DNP eliminates the need for self-attention over the context set by constructing a distance-preserving latent space using a bi-Lipschitz-regularized encoder. At prediction time, each target point performs cross-attention over the context representations, resulting in a total complexity of $\mathcal{O}(NM)$. During training, there is an added cost from the LOBPCG computation. The cost per layer is $\mathcal{O}(Tpq)$ where T is the number of iterations, $p$ and $q$ are the dimensions of the weight matrix.

## A.3   Regression

### A.3.1   Network Architecture

The overall architecture of regression using DNP is as follows.

**Global latent encoder.** The global latent encoder is a 3-layer MLP, constructed similarly to the structure in NP and AttnNP. The input to the MLP is the concatenation of the input variable $\mathbf{x} \in \mathbb{R}^{d_x}$ and output variable $\mathbf{y} \in \mathbb{R}^{d_y}$.

Each context and target pair is first encoded independently, and then the resulting representations are aggregated via the mean function to produce a global summary.

The MLP architecture is as follows:

$$d_x + d_y \to d_h \to d_h \to d_h$$

Note that unless specified, each $\to$ in this text represents a linear layer followed by a Leaky ReLU activation, except for the final layer. The output of the final layer provides an embedding of each input pair $(\mathbf{x}_i, \mathbf{y}_i)$. These embeddings are then aggregated across the entire context set using a mean operation, resulting in a single summary vector. This summary vector is subsequently passed through a separate linear layer to produce the mean and log-variance of a Gaussian distribution over the global latent variable $\mathbf{z}_g$.

$$d_h \to [\mu_g, \Sigma_g] \quad \text{where} \quad \mu_g, \Sigma_g \in \mathbb{R}^{d_z}$$

**Local latent encoder.** To obtain a distortion-bounded latent representation, we first construct a shared 2-layer MLP that maps each input $\mathbf{x}$ into an intermediate feature space of dimension $d_h$. The MLP follows the structure:

$$d_x \to d_h \to d_h$$

The shared backbone is used to extract two components: (i) the distortion-bounded representation $\mathbf{u}$, and (ii) the parameters of the amortized posterior distribution. The distortion-bounded latent representation $\mathbf{u} \in \mathbb{R}^{d_u}$ is obtained by applying a linear projection to the backbone output:

$$d_h \to d_u$$

To obtain the approximate posterior and prior distributions over the local latent variable, the hidden representation from the backbone is concatenated with the corresponding target output $\mathbf{y}$ and passed through an additional linear layer. This layer outputs the mean and log-variance of a Gaussian distribution:

$$d_h + d_y \to [\mu_l, \Sigma_l]$$

where $\mu_l, \Sigma_l \in \mathbb{R}^{d_z}$.

Next, to incorporate context information into each target, we apply a cross-attention between the target's embedding $\mathbf{u}_t$ and all context embeddings $\mathbf{u}_c$ to obtain the attention weights $\alpha_t^c$. These attention weights then aggregate the per-context Gaussian parameters into a target-specific local prior.

**Decoder.** The decoder takes as input the concatenation of the sampled global latent $\mathbf{z}_g \in \mathbb{R}^{d_z}$, the sampled local latent $\mathbf{z}_l \in \mathbb{R}^{d_z}$, and the distortion-bounded embedding $\mathbf{u} \in \mathbb{R}^{d_u}$. This produces a vector of dimension $2d_z + d_u$, which is passed through a 3-layer MLP to output the parameters of the predictive Gaussian over $\mathbf{y} \in \mathbb{R}^{d_y}$:

$$2 * d_z + d_u \to d_h \to d_h \to 2 * d_y$$

**Other baselines.** The network architecture of CNP consists of a 3-layer MLP with hidden dimension $d_h$ and a latent representation of dimension $d_z$. Each context pair $(\mathbf{x}, \mathbf{y})$ is passed through the encoder MLP, and the resulting representations are mean aggregated to form a global representation. The decoder then takes the concatenation of this global representation and the target input $\mathbf{x}_t$ and passes it through another 3-layer MLP to predict the mean and log-variance of the output $\mathbf{y}_t$. Each layer in the MLPs except the output layer is followed by a Leaky ReLU activation.

For NP, we use the architecture proposed in [25]. The encoder is composed of two parallel paths: a deterministic path and a latent path. Both paths utilize a 3-layer MLP with hidden layers of dimension

$d_h$ and an output dimension of $d_z$. Each input-output pair $(\mathbf{x}, \mathbf{y})$ is encoded independently, and the resulting representations are aggregated using a mean operation. In the latent path, the aggregated representation is used to produce the parameters of a multivariate Gaussian distribution, namely the mean $\mu_g$ and log-variance $\log \Sigma_g$ of the global latent variable. The decoder receives the sampled latent variable, the aggregated deterministic representation, and the target input $\mathbf{x}_t$ as input, and outputs the prediction $\mathbf{y}_t$ through a separate 3-layer MLP to predict the mean and log-variance of the output $\mathbf{y}_t$.

For AttnNP and DSVNP, we follow a similar architecture as NP, with the key difference that the mean aggregation in the encoder is replaced by multi-head self-attention and the deterministic path uses a multi-head cross-attention network, as proposed in [25]. The main distinction between AttnNP and DSVNP is that DSVNP models both global and local latent variables by predicting Gaussian distribution parameters for each, where the local latent is conditioned on the global latent. All other components, including the decoder, remain identical to those used in NP.

For both ConvCNP and ConvNP, we follow the architecture proposed in [14, 8], using 1D convolutional layers. The encoder consists of 3 convolutional layers, each with 64 channels and followed by Leaky ReLU activations. The input data is first interpolated onto a uniform grid using a Gaussian kernel before being passed through the encoder. The decoder also consists of 3 convolutional layers, each with 64 channels, and outputs the mean and log-variance of the predictive distribution for each target point.

The models were implemented using PyTorch [43] and trained on an Nvidia GeForce GTX 1080 with 12 GB of RAM. To obtain the singular values for DNP, we use PyTorch's implementation of the LOBPCG method, which employs orthogonal basis selection [51].

### A.3.2   1D Regression

Inputs for both context and target points are drawn uniformly from the interval $[-2, 2]$. The number of context points is sampled uniformly from $[3, 50]$, while the number of target points is fixed at 50. For all kernels, the lengthscale and the output scale values are sampled uniformly from $[0.6, 1.0]$ and $[0.1, 1.0)$, respectively. Additionally, for the Periodic kernel, the period parameter is sampled uniformly from the interval $[0.5, 1.5)$. A likelihood noise of $0.02$ is added for all the cases. A batch size of 50 is used for training. During evaluation, the number of context and target points is sampled from the same distributions used during training. The models used the following hyperparameters: the encoder/decoder hidden dimension was set to $d_h = 64$, the latent dimension to $d_z = 64$, and the distance-aware latent dimension to $d_u = 64$. All the models are trained using the Adam optimizer [26] at a learning rate of $1e^{-3}$, for 200 epochs. For DNP, the bi-Lipschitz constraint is enforced with lower and upper singular value bounds $\lambda_1 = 0.1, \lambda_2 = 1.0$, and the balancing the ELBO and bi-Lipschitz regularization is set to $\beta = 1$.

### A.3.3   Synthetic to Real-World Regression

The data generation process follows [14]. For each training batch, the number of context points is sampled uniformly from the interval $[3, 50]$, while the number of target points is fixed at 50. At test time, 30 context points are randomly selected from the train data. The following hyperparameters were used for the models: encoder/decoder hidden dimension $d_h = 64$, latent dimension $d_z = 64$, and distance-aware latent dimension $d_u = 64$. All networks were trained with the Adam optimizer [26] at a learning rate of $1e^{-3}$, using a batch size of 50 for 200 epochs. For DNP, the lower and upper bounds on the singular values were set to $\lambda_1 = 0.1, \lambda_2 = 1.0$, with a trade-off weight $\beta = 1$ balancing the ELBO and bi-Lipschitz regularization.

### A.3.4   Multi-output Real-World Regression

We evaluate DNP on three benchmark datasets: SARCOS [61], which models the inverse dynamics of a seven-degree-of-freedom robot arm by predicting seven joint torques from 21 inputs (positions, velocities, accelerations); Water Quality (WQ) [7], which predicts species abundances in Slovenian rivers from 16 physical and chemical indicators; and SCM20D [50], a supply-chain time series for multi-item demand forecasting. All datasets are standardized to zero mean and unit variance per feature and split 80/20 into training and test sets.

For every training batch, the number of context points is randomly sampled from a uniform distribution in the range $[3, 100]$, and the number of target points is 100. The network architecture uses hidden and latent dimensions $d_h = 128, d_z = 128$, and the distance-aware latent dimension $d_u = 128$. Models are trained using Adam optimizer [26] with a learning rate of $1e^{-3}$, and a batch size of 100, for 100 epochs. At test time, we randomly select 30 context points from the train data. For the bi-Lipschitz regularizer, we use $\lambda_1 = 0.1, \lambda_2 = 1.0$ and set the ELBO/bi-Lipschitz trade-off weight to $\beta = 1$.

## A.4 Image Classification and OOD Detection

For classification tasks on CIFAR-10 and CIFAR-100, we use a VGG-16 network [48] with Leaky ReLU activation as the feature extractor.

**Global latent encoder.** In the global latent encoder, the feature embedding obtained from VGG-16 is concatenated with the one-hot encoded label $y$ to form the input representation. The flow of operations is summarized as:

$$d_x \xrightarrow{\text{VGG-16}} d_h$$
$$[d_h, \text{one\_hot}(y)] \to d_g \to [\mu_g, \Sigma_g] \quad \text{where} \quad \mu_g, \Sigma_g \in \mathbb{R}^{d_z}$$

**Local latent encoder.** The local latent encoder also uses the feature embedding extracted by VGG-16. Each input $\mathbf{x}$ is passed through VGG-16 to obtain an embedding of size $d_h$:

$$d_x \xrightarrow{\text{VGG-16}} d_h$$

This backbone is used to produce two outputs: (1) A distortion-bounded latent embedding $\mathbf{u} \in \mathbb{R}^{d_u}$, obtained via a linear projection:

$$d_h \to d_u$$

(2) The parameters (mean and variance) of the amortized posterior distribution over the local latent variable:

$$d_h + d_y \to [\mu_l, \Sigma_l] \quad \text{where} \quad \mu_l, \Sigma_l \in \mathbb{R}^{d_z}$$

Cross-attention is then applied between the distortion-bounded target embedding $\mathbf{u}_t$ and the set of context embeddings $\mathbf{u}_c$ to obtain an attention-based aggregation of local latent parameters.

**Decoder.** The decoder takes as input the concatenation of the sampled global latent variable $\mathbf{z}_g \in \mathbb{R}^{d_z}$, the sampled local latent variable $\mathbf{z}_t \in \mathbb{R}^{d_z}$, and the distortion-bounded latent embedding $\mathbf{u} \in \mathbb{R}^{d_u}$. The resulting vector of dimension $2d_z + d_u$ is passed through a 3-layer MLP to predict the parameters of a categorical distribution over the output classes.

$$2d_z + d_u \to d_h \to d_h \to d_y$$

The other baselines follow a similar architecture to that described in Section A.3.1, with the key difference that a VGG-16 network is used as a feature extractor for the input images. The extracted feature embedding is concatenated with the one-hot encoded label vector before being passed through the subsequent layers. Similar to DNP, the predictive distribution is modeled as a categorical distribution over the output classes.

For the experiments, the networks are trained on the training sets of CIFAR-10 and CIFAR-100, each consisting of 60,000 images categorized into 10 and 100 classes, respectively. Evaluation is performed on the corresponding test sets, each containing 10,000 images. The images have a dimension of $32 \times 32$, and are normalized to zeros mean and unit variance. For OOD detection, images from the test set of the SVHN, CIFAR100/CIFAR10 and the TinyImageNet dataset is used. For a fair comparison, all the OOD images are resized to $32 \times 32$ and are normalized the same way as the CIFAR10/CIFAR100 images. For training, the number of context points is randomly chosen between from a uniform distribution between 16 and 128. A batch size of 128 is used. The network architecture has the dimesnions $d_h, d_z = 512$ and $d_u = 256$. The models were implemented using PyTorch [43] and trained on an NVIDIA A100 GPU with 40 GB of memory. Models are trained using the Adam optimizer [26] with a learning rate of $1e^{-4}$ for 200 epochs. At test time, 100 context points are randomly selected from the train set. For the bi-Lipschitz regularizer, $\lambda_1 = 0.1$ and

Table 8: Performance comparison across different kernel functions for 1D synthetic regression for context set. Results are averaged over 10 runs.

| Method | RBF | | Matérn-$\frac{5}{2}$ | | Periodic | | Latency ($\downarrow$) |
| | LL ($\uparrow$) | ECE ($\downarrow$) | LL ($\uparrow$) | ECE ($\downarrow$) | LL ($\uparrow$) | ECE ($\downarrow$) | (ms) |
|---|---|---|---|---|---|---|---|
| CNP [11] | 0.832±0.003 | 0.072±0.034 | 0.828±0.003 | 0.073±0.047 | 0.332±0.004 | 0.139±0.043 | **0.046** |
| NP [12] | 0.856±0.002 | 0.084±0.034 | 0.692±0.004 | 0.093±0.037 | 0.631±0.003 | 0.115±0.040 | 0.095 |
| ConvCNP [14] | 0.855±0.002 | 0.088±0.032 | 0.926±0.002 | 0.094±0.038 | 0.438±0.002 | 0.119±0.030 | 0.058 |
| ConvNP [8] | 0.964±0.004 | 0.092±0.052 | 0.920±0.002 | 0.062±0.013 | 0.766±0.003 | 0.121±0.021 | 0.117 |
| AttnNP [25] | 1.313±0.002 | 0.076±0.018 | 0.931±0.006 | 0.077±0.011 | 0.764±0.001 | 0.072±0.012 | 0.106 |
| DSVNP [63] | 1.322±0.006 | 0.082±0.027 | 1.018±0.003 | 0.065±0.018 | 0.829±0.001 | 0.102±0.023 | 0.109 |
| Ours | **1.439±0.004** | **0.064±0.021** | **1.229±0.001** | **0.054±0.012** | **0.934±0.001** | **0.068±0.022** | 0.102 |

$\lambda_2 = 1.0$, with the ELBO/bi-Lipschitz trade-off weight $\beta = 0.5$. To obtain the singular values, we use PyTorch's implementation of the LOBPCG method, which employs orthogonal basis selection [51].

# B  Additional Experiments

## B.1  1D Synthetic Regression

Table 2 reports the log-likelihood (LL) and expected calibration error (ECE) for the target points, excluding the context data. In Table 8, following the evaluation protocol of [25], we provide the LL and ECE scores on the context points for the 1D synthetic regression task. Consistent with the results on the target set, DNP achieves higher LL and lower ECE scores, indicating better predictive performance and calibration. In terms of inference speed, CNP incurs the lowest computational cost due to its fully deterministic architecture. In contrast, ConvNP is more expensive because of its convolutional structure. Meanwhile, AttnNP, DSVNP, and DNP exhibit comparable inference times, as they share similar architectural components such as attention and latent variable sampling.

To assess the robustness of DNP to observation noise, we evaluate the performance of the NP models on 1D synthetic regression tasks. The functions are generated from the RBF kernel, with noise levels of 5%, 10%, and 15% on the observations. Evaluation on the target set excluding the context data is provided in Table 9. Results show that DNP consistently achieves the highest log-likelihood across all noise settings, while also maintaining strong uncertainty calibration with the lowest ECE in most cases. We also study the role of geometry-aware regularization. Table 10 reports the effect of bi-Lipschitz regularization on ConvCNP and ConvNP across different kernel families in the 1D synthetic regression task. Incorporating the bi-Lipschitz constraint consistently improves log-likelihood and reduces calibration error compared to the unregularized counterparts, with the strongest gains observed for ConvNP.

Table 9: Performance comparison for 1D synthetic regression under noisy data for RBF kernel function. Results are averaged over 10 runs.

| Method | 5% Noise | | 10% Noise | | 15% Noise | |
| | LL ($\uparrow$) | ECE ($\downarrow$) | LL ($\uparrow$) | ECE ($\downarrow$) | LL ($\uparrow$) | ECE ($\downarrow$) |
|---|---|---|---|---|---|---|
| CNP [11] | 0.462±0.003 | 0.117±0.032 | 0.421±0.006 | 0.104±0.028 | 0.321±0.003 | 0.073±0.031 |
| NP [12] | 0.475±0.005 | 0.112±0.043 | 0.436±0.012 | 0.094±0.031 | 0.364±0.004 | 0.095±0.021 |
| ConvCNP [14] | 0.512±0.005 | 0.090±0.048 | 0.502±0.001 | 0.083±0.029 | 0.452±0.004 | 0.054±0.027 |
| ConvNP [8] | 0.532±0.004 | 0.111±0.071 | 0.526±0.003 | 0.082±0.039 | 0.479±0.005 | 0.051±0.019 |
| AttnNP [25] | 0.571±0.004 | 0.082±0.031 | 0.515±0.011 | 0.097±0.010 | 0.485±0.001 | **0.021±0.026** |
| DSVNP [63] | 0.567±0.003 | 0.094±0.043 | 0.532±0.008 | 0.092±0.011 | 0.476±0.005 | 0.048±0.038 |
| Ours | **0.625±0.004** | **0.073±0.036** | **0.585±0.003** | **0.081±0.018** | **0.522±0.007** | 0.037±0.029 |

## B.2  Ablation Study

Table 11 presents an ablation study evaluating the impact of the number of context points on CIFAR-10 (ID) vs. CIFAR-100 (OOD) classification performance for both AttnNP and DNP. We observe that the accuracy and AUPR scores are robust to the number of context points for both models, remaining approximately constant across different context set sizes, with only minor fluctuations in ECE. Notably, DNP consistently outperforms AttnNP across all metrics and context sizes, achieving

Table 10: Evaluation of bi-Lipschitz regularization on ConvNP for 1D synthetic regression. Results are averaged over 10 runs.

| Method | RBF | | Matérn-$\frac{5}{2}$ | | Periodic | |
|---|---|---|---|---|---|---|
| | LL ($\uparrow$) | ECE ($\downarrow$) | LL ($\uparrow$) | ECE ($\downarrow$) | LL ($\uparrow$) | ECE ($\downarrow$) |
| ConvCNP | 0.496±0.007 | 0.118±0.054 | 0.329±0.001 | 0.098±0.032 | 0.335±0.002 | 0.106±0.032 |
| ConvNP | 0.609±0.005 | 0.190±0.074 | 0.313±0.002 | 0.137±0.095 | 0.431±0.003 | 0.117±0.011 |
| ConvCNP + bi-Lipschitz | 0.507±0.012 | 0.086±0.013 | 0.400±0.014 | 0.081±0.014 | 0.399±0.004 | 0.082±0.021 |
| ConvNP + bi-Lipschitz | 0.702±0.008 | 0.073±0.018 | 0.462±0.016 | 0.093±0.019 | 0.515±0.003 | 0.098±0.018 |

superior calibration and predictive performance. In line with this, we use 100 context points for the main experiments, as it offers a good balance between performance and computational cost.

Table 11: Ablation study on the number of context points for CIFAR-10 vs CIFAR-100 classification. Results are averaged over 10 runs.

| # Context | AttnNP | | | DNP | | |
|---|---|---|---|---|---|---|
| | Accuracy ($\uparrow$) | ECE ($\downarrow$) | AUPR ($\uparrow$) | Accuracy ($\uparrow$) | ECE ($\downarrow$) | AUPR ($\uparrow$) |
| 50 | 90.36±0.01 | 0.021±0.001 | 87.12±0.01 | 91.92±0.01 | 0.015±0.002 | 90.51±0.01 |
| 100 | 90.37±0.01 | 0.020±0.000 | 87.11±0.01 | 91.97±0.01 | 0.011±0.002 | 90.58±0.01 |
| 200 | 90.39±0.01 | 0.021±0.003 | 87.18±0.02 | 91.96±0.02 | 0.012±0.002 | 90.58±0.02 |
| 500 | 90.40±0.01 | 0.023±0.002 | 87.17±0.02 | 91.95±0.01 | 0.010±0.001 | 90.58±0.02 |

Table 12 reports the metrics on the effect of varying the distance-aware latent dimension $d_u$ and latent dimension $d_z$ on CIFAR-10 (ID) vs. CIFAR-100 (OOD) classification. We observe that increasing both $d_u$ and $d_z$ generally improves performance across all metrics, with the best results achieved when $d_u = 256$ and $d_z = 512$.

Table 12: Ablation study on the impact of varying $d_u$ and $d_z$ on classification performance for CIFAR-10 vs. CIFAR-100. Results are averaged over 10 runs.

| $d_u, d_z$ | Accuarcy($\uparrow$) | ECE($\downarrow$) | AUPR($\uparrow$) |
|---|---|---|---|
| 512, 512 | 91.36±0.02 | 0.011±0.001 | 90.14±0.01 |
| 512, 256 | 90.47±0.02 | 0.014±0.000 | 90.39±0.01 |
| 256, 512 | **91.97±0.01** | **0.011±0.002** | **90.58±0.01** |
| 256, 256 | 90.64±0.01 | 0.012±0.002 | 90.36±0.01 |
| 256, 128 | 90.82±0.01 | 0.014±0.002 | 89.52±0.01 |
| 128, 256 | 90.44±0.01 | 0.014±0.001 | 90.04±0.01 |
| 128, 128 | 88.83±0.01 | 0.019±0.001 | 87.81±0.01 |

Table 13 presents an ablation study examining the impact of varying the lower ($\lambda_1$) and upper ($\lambda_2$) bounds of the bi-Lipschitz constraint on model performance. The configuration $\lambda_1 = 0.1$ and $\lambda_2 = 1.0$ yields the best performance, balancing regularization strength and representational flexibility. As $\lambda_2$ increases while keeping $\lambda_1 = 0.1$, we observe a modest trade-off: the accuracy remains competitive, but there is a slight degradation in calibration (ECE) and AUPR scores as $\lambda_2$ becomes larger. In contrast, increasing $\lambda_1$ (e.g., to 0.6 or higher) or setting $\lambda_1 = \lambda_2$ results in more substantial performance degradation. This suggests that overly tight constraints on the bi-Lipschitz regularizer hinder the model's ability to adapt to complex input-output relationships, limiting both its expressiveness and the quality of its uncertainty estimates.

Table 14 presents an ablation study on the effect of the trade-off parameter $\beta$ which balances the ELBO and the bi-Lipschitz regularization loss, on CIFAR-10 (ID) vs. CIFAR-100 (OOD) classification. As $\beta$ increases, calibration and uncertainty quantification improve significantly, peaking around $\beta = 0.5$. Beyond this point, calibration remains stable but accuracy slightly drops, suggesting that excessive regularization limits predictive performance. While $\beta$ was manually tuned in this work to balance predictive performance and uncertainty calibration, automating its selection remains an important direction for future work [24].

Table 13: Effect of bi-Lipschitz constraint bounds ($\lambda_1$, $\lambda_2$) on CIFAR10 vs CIFAR100 classification. Results are averaged over 10 runs.

| $\lambda_1$, $\lambda_2$ | Accuarcy($\uparrow$) | ECE($\downarrow$) | AUPR($\uparrow$) | $\lambda_1$, $\lambda_2$ | Accuarcy($\uparrow$) | ECE($\downarrow$) | AUPR($\uparrow$) |
|---|---|---|---|---|---|---|---|
| 0.1, 0.6 | 89.61±0.01 | 0.014±0.002 | 88.98±0.01 | 0.2, 1.0 | 91.41±0.01 | 0.013±0.001 | 89.72±0.01 |
| 0.1, 0.8 | 91.09±0.01 | 0.012±0.001 | 89.46±0.02 | 0.4, 1.0 | 90.89±0.01 | 0.012±0.002 | 88.43±0.01 |
| 0.1, 1.0 | 91.97±0.01 | 0.011±0.002 | 90.58±0.01 | 0.6, 1.0 | 89.60±0.01 | 0.017±0.002 | 88.92±0.02 |
| 0.1, 1.2 | 91.92±0.01 | 0.011±0.002 | 89.36±0.01 | 0.8, 1.0 | 89.86±0.01 | 0.027±0.003 | 89.05±0.01 |
| 0.1, 1.4 | 92.02±0.01 | 0.015±0.002 | 89.52±0.01 | 1.0, 1.0 | 89.92±0.01 | 0.031±0.03 | 88.73±0.01 |

Table 14: Evaluation of the trade-off parameter between the ELBO and the bi-Lipschitz regularization loss on CIFAR10 vs CIFAR100 classification. Results are averaged over 10 runs.

| $\beta$ | Accuracy ($\uparrow$) | ECE ($\downarrow$) | AUPR ($\uparrow$) |
|---|---|---|---|
| 0 | **92.08±0.01** | 0.042±0.001 | 87.85±0.01 |
| 0.1 | 92.01±0.01 | 0.023±0.003 | 88.14±0.01 |
| 0.2 | 91.92±0.01 | 0.019±0.001 | 89.62±0.01 |
| 0.5 | 91.97±0.01 | **0.011±0.002** | **90.58±0.01** |
| 0.7 | 90.62±0.01 | 0.011±0.003 | 90.42±0.02 |
| 1.0 | 90.59±0.01 | 0.012±0.002 | 90.43±0.01 |

Table 15: Comparison of the Laplace and dot-product similarity functions for cross-attention on CIFAR10 (ID) vs CIFAR100 (OOD). Results are averaged over 10 runs.

| Attention Type | Accuracy($\uparrow$) | ECE($\downarrow$) | AUPR($\uparrow$) |
|---|---|---|---|
| Laplace | **91.97±0.01** | **0.011±0.002** | **90.58±0.01** |
| Dot-product | 91.12±0.01 | 0.018±0.002 | 89.62±0.01 |

We also evaluate the performance when using Laplacian attention (Eq. 5) and dot-product attention, given by

$$\alpha_t^c = \frac{\exp\left(\frac{\mathbf{u}_t^T \mathbf{u}_c}{\sqrt{d_u}}\right)}{\sum_{c' \in C} \exp\left(\frac{\mathbf{u}_t^T \mathbf{u}_{c'}}{\sqrt{d_u}}\right)}. \tag{13}$$

The results are presented in Table 15. While both attention mechanisms perform similarly overall, Laplace attention yields improved classification accuracy, better uncertainty calibration, and OOD detection.

Table 16 presents an ablation analysis of the global and local latent paths in the DNP model. We observe that using only the local latent path yields the best performance in terms of classification accuracy and calibration (ECE), indicating its strong contribution to modeling instance-specific uncertainty. In contrast, the global latent path provides slightly better AUPR. Incorporating both paths yields the best overall performance (see main results in Table 5).

Table 16: Ablation study evaluating the contribution of global and local latent paths in DNP.

| Latent path | Accuracy ($\uparrow$) | ECE ($\downarrow$) | AUPR ($\uparrow$) |
|---|---|---|---|
| DNP (Global latent only) | 90.44±0.01 | 0.046±0.002 | **90.59±0.01** |
| DNP (Local latent only) | **91.11±0.01** | **0.016±0.002** | 90.56±0.01 |

## C Proof of Proposition 1

To show that the generative model defined in Eq. 7 is a stochastic process, it needs to satisfy two conditions of the Kolmogorov Extension Theorem [41]:

1. **Permutation Invariance (Exchangeability):** The joint distribution $p_{x_{1:N}}(y_{1:N})$ must be invariant under any permutation $\pi$ of the indices $\{1, \ldots, N\}$; that is,

$$p_{x_{1:N}}(y_{1:N}) = p_{x_{\pi(1:N)}}(y_{\pi(1:N)}) \tag{14}$$

2. **Consistency (Marginalization):** For any $N$ and any $M < N$, the marginal distribution over any subset $\{y_1, \ldots, y_M\}$ of the outputs must be recoverable by integrating out the remaining variables from the joint distribution $p_{x_{1:N}}(y_{1:N})$.

We now prove each of these in turn. For the proof, we assume that the joint distribution over targets, latents, and inputs factorizes as

$$p_{\mathbf{x}_{1:N}}(\mathbf{y}_{1:N}) = \iint \prod_{i=1}^{N} p(\mathbf{y}_i | \mathbf{z}_G, \mathbf{z}_i, \mathbf{x}_i) \, p(\mathbf{z}_i | \mathbf{x}_i) \, p(\mathbf{z}_G) \, d\mathbf{z}_{1:N} \, d\mathbf{z}_G \tag{15}$$

**Exchangeability:** Let $\pi$ be any permutation of $\{1, 2, \cdots, N\}$. Since we use a permutation-invariant distribution over the global and local latent variables i.e., the prior $p(\mathbf{z}_G)$ is shared across all inputs and each $p(\mathbf{z}_i | \mathbf{x}_i)$ is independently defined—the joint distribution remains invariant under reordering of the inputs and outputs:

$$\begin{aligned} p_{\mathbf{x}_{1:N}}(\mathbf{y}_{1:N}) &= \iint \prod_{i=1}^{N} p(\mathbf{y}_i \mid \mathbf{z}_G, \mathbf{z}_i, \mathbf{x}_i) \, p(\mathbf{z}_i \mid \mathbf{x}_i) \, p(\mathbf{z}_G) \, d\mathbf{z}_{1:N} \, d\mathbf{z}_G \\ &= \iint \prod_{i=1}^{N} p(\mathbf{y}_{\pi(i)} \mid \mathbf{z}_G, \mathbf{z}_{\pi(i)}, \mathbf{x}_{\pi(i)}) \, p(\mathbf{z}_{\pi(i)} \mid \mathbf{x}_{\pi(i)}) \, p(\mathbf{z}_G) \, d\mathbf{z}_{1:N} \, d\mathbf{z}_G \\ &= p_{\mathbf{x}_{\pi(1:N)}}(\mathbf{y}_{\pi(1:N)}) \end{aligned} \tag{16}$$

Thus, the model defines a permutation-invariant distribution over the input-output pairs, satisfying the exchangeability condition required by the Kolmogorov Extension Theorem.

**Consistency:** From Eq. (7), we have

$$\int p_{\mathbf{x}_{1:N}}(\mathbf{y}_{1:N}) \, d\mathbf{y}_{M+1:N} = \int \left[ \iint \prod_{i=1}^{N} p(\mathbf{y}_i \mid \mathbf{z}_G, \mathbf{z}_i, \mathbf{x}_i) \, p(\mathbf{z}_i \mid \mathbf{x}_i) \, p(\mathbf{z}_G) \, d\mathbf{z}_{1:N} \, d\mathbf{z}_G \right] d\mathbf{y}_{M+1:N}. \tag{17}$$

By Fubini–Tonelli [44] (since all densities are nonnegative and integrable), we may interchange the integrals and the product over $i$, yielding

$$\begin{aligned} \int p_{\mathbf{x}_{1:N}}(\mathbf{y}_{1:N}) \, d\mathbf{y}_{M+1:N} = \iint &\left[ \int \prod_{i=M+1}^{N} p(\mathbf{y}_i \mid \mathbf{z}_G, \mathbf{z}_i, \mathbf{x}_i) \, p(\mathbf{z}_i \mid \mathbf{x}_i) \, d\mathbf{y}_{M+1:N} \, d\mathbf{z}_{M+1:N} \right] \\ &\times \prod_{i=1}^{M} p(\mathbf{y}_i \mid \mathbf{z}_G, \mathbf{z}_i, \mathbf{x}_i) \, p(\mathbf{z}_i \mid \mathbf{x}_i) \, p(\mathbf{z}_G) \, d\mathbf{z}_{1:M} \, d\mathbf{z}_G. \end{aligned} \tag{18}$$

The inner bracket integrates to 1 (marginalizing out the last $N - M$ points), leaving exactly

$$\iint \prod_{i=1}^{M} p(\mathbf{y}_i \mid \mathbf{z}_G, \mathbf{z}_i, \mathbf{x}_i) \, p(\mathbf{z}_i \mid \mathbf{x}_i) \, p(\mathbf{z}_G) \, d\mathbf{z}_{1:M} \, d\mathbf{z}_G, \tag{19}$$

which matches the form of $p_{\mathbf{x}_{1:M}}(\mathbf{y}_{1:M})$. Hence, marginal consistency holds. $\square$

# D Derivation of ELBO Loss

The generative model in Eq. 7 is intractable, we introduce a variational approximation $q_\phi(\mathbf{z}_t, \mathbf{z}_G)$ to approximate the true posterior. Using the Evidence Lower Bound (ELBO) [27], we rewrite the log-marginal likelihood by multiplying and dividing by $q_\phi(\mathbf{z}_t, \mathbf{z}_G)$:

$$\log p(\mathbf{y}_t | \mathbf{x}_t, \mathbf{x}_C, \mathbf{y}_C) = \log \iint p(\mathbf{y}_t, \mathbf{z}_t, \mathbf{z}_G | \mathbf{x}_t, \mathbf{x}_C, \mathbf{y}_C) \frac{q_\phi(\mathbf{z}_t, \mathbf{z}_G)}{q_\phi(\mathbf{z}_t, \mathbf{z}_G)} \, d\mathbf{z}_t \, d\mathbf{z}_G. \tag{20}$$

Applying Jensen's inequality, we obtain:

$$\log p(\mathbf{y}_t | \mathbf{x}_t, \mathbf{x}_C, \mathbf{y}_C) \geq \mathbb{E}_{q_\phi} \left[ \log \frac{p(\mathbf{y}_t, \mathbf{z}_t, \mathbf{z}_G | \mathbf{x}_t, \mathbf{x}_C, \mathbf{y}_C)}{q_\phi(\mathbf{z}_t, \mathbf{z}_G)} \right]. \tag{21}$$

This expectation defines the ELBO.

To make inference tractable, we assume a structured variational approximation where the posterior factorizes as:

$$q_\phi(\mathbf{z}_t, \mathbf{z}_G) = q_{\phi_G}(\mathbf{z}_G|\mathbf{x}_T, \mathbf{y}_T)q_{\phi_L}(\mathbf{z}_t|\mathbf{x}_t, \mathbf{y}_t, \mathbf{x}_C, \mathbf{y}_C). \tag{22}$$

Expanding the ELBO by substituting the factorization, we obtain:

$$\log p(\mathbf{y}_t|\mathbf{x}_t, \mathbf{x}_C, \mathbf{y}_C) \geq \mathbb{E}_{q_\phi}[\log p(\mathbf{y}_t|\mathbf{z}_G, \mathbf{z}_t, \mathbf{x}_t) + \log p(\mathbf{z}_t|\mathbf{x}_t, \mathbf{x}_C, \mathbf{y}_C) + \log p(\mathbf{z}_G|\mathbf{x}_C, \mathbf{y}_C)]$$
$$-\mathbb{E}_{q_\phi}\left[\log q_{\phi_L}(\mathbf{z}_t|\mathbf{x}_t, \mathbf{y}_t, \mathbf{x}_C, \mathbf{y}_C) + \log q_{\phi_G}(\mathbf{z}_G|\mathbf{x}_T, \mathbf{y}_T)\right]. \tag{23}$$

Rewriting the expectation terms in terms of Kullback-Leibler (KL) divergences, we obtain the final ELBO:

$$\log p(\mathbf{y}_t|\mathbf{x}_t, \mathbf{x}_C, \mathbf{y}_C) \geq \mathbb{E}_{q_{\phi_G}q_{\phi_L}}\left[\log p(\mathbf{y}_t|\mathbf{z}_G, \mathbf{z}_t, \mathbf{x}_t)\right]$$
$$-D_{\mathrm{KL}}(q_{\phi_L}(\mathbf{z}_t|\mathbf{x}_t, \mathbf{y}_t, \mathbf{x}_C, \mathbf{y}_C) \parallel p_{\theta_L}(\mathbf{z}_t|\mathbf{x}_t, \mathbf{x}_C, \mathbf{y}_C)) - D_{\mathrm{KL}}(q_{\phi_G}(\mathbf{z}_G|\mathbf{x}_T, \mathbf{y}_T) \parallel p_{\theta_G}(\mathbf{z}_G|\mathbf{x}_C, \mathbf{y}_C)) \tag{24}$$

This ELBO consists of: 1. A reconstruction term that encourages accurate predictions of $\mathbf{y}_t$. 2. Two KL divergence regularization terms that enforce approximate posteriors $q_{\phi_G}$ and $q_{\phi_L}$ to be close to their respective priors.

