# OpenReview forum: "Distance-informed Neural Processes"
_NeurIPS.cc/2025/Conference — NeurIPS 2025 poster_

### Official Review · Reviewer_jh3q · 2025-06-17

**Clarity:** 3
**Significance:** 3
**Originality:** 3
**Rating:** 5
**Confidence:** 3

**Summary:**

The paper introduces so called Distance-informed Neural Processes (DNP). DNPs build on Neural Processes but additionally include a local latent path and variable, with the main objective of an improved uncertainty calibration. To this end, bi-Lipschitz regularization is used to preserve distances from input to output. DNPs are evaluated on different rasks ranging from onedimensional regression to image classification, with the novel method typically showing strong accuracy and improved uncertainty calibration compared to competing methods such as Attentive NPs.

**Questions:**

1. The paper on ConvNP (Foong et al. 2020) states to propose a new maximum-likelihood objective replacing ELBO, how does the objective in the paper (cf. Appendix D) compare to it?
2. It was surprising to see DNP being so overconfident in some settings (cf. Figure 3). What are the drivers behind this?
3. Is there an alternative to the LOBPCG method applied?
If these questions are addressed satisfactorily, I may increase my evaluation score.

**Ethical Concerns:**

["NO or VERY MINOR ethics concerns only"]

**Final Justification:**

This is a paper of high quality with high significance. It showcases strong results across a variety of experiments from different contexts. The clarity of the paper was further improved during the discussion, and the questions have been answered satisfactorily. I thus recommend to accept the paper.

**Limitations:**

yes

**Paper Formatting Concerns:**

I have not noticed any major formatting issues in this paper. However, the citation formatting does not seem to follow NeurIPS standards.

**Quality:**

4

**Strengths And Weaknesses:**

Quality: I was not able to identify a technical issue in the paper. A proof for Proposition 1 is provided in Appendix C. A strong point is that a variety of applications/experiments from different contexts is shown, including an illustration of predictive distributions/confidence intervals. In addition, substantial additional material is contained in the Appendix, and the code is available in the supplementary material. Overall, I believe the paper to be a thoughtful and sound work.

Clarity: The paper was quite clear and relatively straightforward to read. The motivation for the research is presented well, as is the justification of the bi-Lipschitz regularization. Most of the technicals are also presented in a straightforward manner. Much detailed information is provided. I have only smaller suggestions for improvement. What was most confusing to me was the description of Table 4/5 and the corresponding results. The differences between using the two different datasets as ID should be discussed properly. Also, why is latency only shown for the CIFAR10-case? Moreover, I find it suboptimal that some quite closely related work is discussed in Section 2 “Background”, while some is discussed in Section 4 “Related Works”. For example, I was aware of the existence of ConvNP (Foong et al. 2020) and was thus surprised to not see it being mentioned in Section 2, but then later found it to be mentioned in Section 4. Another smaller point is that I did not get the rationale behind sampling the number of context points in Section 5.2.

Significance: Machine learning methods which provide well-calibrated uncertainty estimates are, without doubt, highly valuable. I thus believe that the paper aims at an important contribution and it improves upon existing NPs. That being said, Figure 3, especially the cases (c) and (d), show that the suggested method DNP still can suffer from substantially underestimating uncertainty and thus yield estimations that are way too confident, even slightly OOD. This would warrant further discussion.

Originality: To the best of my knowledge, while bi-Lipschitz regularization has been applied to many variants of neural networks, applying bi-Lipschitz regularization to NPs is novel and promising. I also found the reasoning for originality to be convincing overall.

In sum, I found the paper to be an interesting read and wish the authors all the best for their work on uncertainty calibration.

Reference: Foong, A., Bruinsma, W., Gordon, J., Dubois, Y., Requeima, J., & Turner, R. (2020). Meta-learning stationary stochastic process prediction with convolutional neural processes. Advances in Neural Information Processing Systems, 33, 8284-8295.

---

> ### Author Rebuttal · Authors · 2025-07-30
>
> We thank you for the positive and constructive feedback. We are glad that the clarity, technical soundness, and significance of our work were recognized. Below, we address the specific questions and suggestions raised.
>
> **Clarifications on Tables 4 and 5**
>
> Thank you for pointing this out. We will add the distinction between ID and OOD datasets in Table 4 and Table 5. Additionally, latency was reported only for the CIFAR10-ID setting as a representative case. The CIFAR100 experiment uses the same model architecture, with the only difference being the number of output classes (10 vs. 100), which had a marginal impact on inference speed. We will also include the following latency results for the CIFAR100 in Table 5 setting for completeness.
>
> | Model         | Latency (ms)        |
> |---------------|---------------------|
> | Deterministic | 3.662 ± 0.110       |
> | CNP           | 3.825 ± 0.116       |
> | NP            | 6.228 ± 0.117       |
> | AttnNP        | 8.424 ± 0.117       |
> | DSVNP         | 8.637 ± 0.125       |
> | Ours          | 6.876 ± 0.118       |
>
> **Organization of Related Work**
>
> Section 2 was primarily intended to provide background for readers less familiar with NPs, focusing on the core models CNP, NP, and AttnNP that our method builds upon. Due to space constraints, we chose to defer the discussion of more recent or orthogonal variants such as ConvNP and DSVNP to Section 4.
>
> **Sampling the Number of Context Points**
>
> We sample the number of context points per task to reflect the meta-learning setting, where the model must generalize to tasks with varying context sizes. This sampling strategy improves robustness and better reflects the uncertainty and variability encountered in real-world tasks. Moreover, this is a standard procedure followed in most NP literature, including NP, AttnNP, and ConvNP, to ensure fair comparison and realistic evaluation.
>
> **Q1. ConvNP Objective function**
>
> The ML-based objective in ConvNP involves sampling a latent variable $z \sim p(z|D_c)$ from a ConvCNP encoder and marginalizing over it, without learning a posterior conditioned on the target set. This approach is effective for models focused purely on predictive performance and is simpler to train. However, it does not provide the explicit control over the latent space structure that our distance-aware design requires.
> In our case, the use of a VI framework is essential due to the way we introduce inductive bias into the model. The local latent variable is designed to encode the geometry of the input structure. To effectively model and learn from this structure, it is crucial to maintain a posterior distribution over the latent variable that adapts to the task. The VI framework allows us to learn this posterior in an amortized manner and is therefore better aligned with the goals of the model than the ML-based objective of ConvNP.
>
> **Q2. Underestimation of Uncertainty**
>
> This is a good observation. The underlying reason is that the bi-Lipschitz constraint does not strictly preserve input geometry; it only enforces approximate distance preservation within a bounded range. As a result, the model may sometimes underestimate distances between some inputs in the latent space, which can lead to overconfident predictions in certain regions.
> This reflects an inherent trade-off: enforcing geometry too rigidly may compromise the model’s expressiveness, making it less able to adapt to complex or non-uniform data distributions. To manage this, we allow bounded flexibility in the Lipschitz constants, which provides a balance between geometric regularization and functional capacity. This trade-off and its impact on uncertainty are further explored in our ablations (see Tables 6 and 11).
>
> **Q3. Alternatives to LOBPCG**
>
> An alternative to LOBPCG is the Lanczos algorithm, which is another iterative method for estimating the extreme eigenvalues of symmetric matrices. We considered this option. However, in our experiments, we found LOBPCG to offer more stable and accurate estimates and decided to use LOBPCG.

---

> > ### Comment · Reviewer_jh3q · 2025-08-01
> > **Answer to Rebuttal**
> >
> > I have carefully read all reviews as well as the author responses. Thank you very much for clarifying my questions and your suggested updates to the paper. I have no further questions at this time and still find the paper to be worthy of acceptance.

---

### Official Review · Reviewer_HQey · 2025-06-22

**Clarity:** 3
**Significance:** 3
**Originality:** 2
**Rating:** 4
**Confidence:** 4

**Summary:**

The paper introduces Distance-informed Neural Processes (DNP), a novel variant of Neural Processes that improves uncertainty caliration and OOD detection by integrating both global and distance-aware local latent variables. DNP employs bi-Lipschitz regularization on its local latent space to preserve geometric structure, enabling better similarity modeling. Experiments show state-of-the-art results in regression and classification tasks with improved uncertainty calibration.

**Questions:**

See the weakness part.

**Ethical Concerns:**

["NO or VERY MINOR ethics concerns only"]

**Final Justification:**

The authors mostly addressed my concerns from more experiments and discussions.

**Limitations:**

See the weakness part.

**Paper Formatting Concerns:**

No concerns

**Quality:**

2

**Strengths And Weaknesses:**

I summarize the strength of this work as follows:

(1) This paper is easy to follow and the introduction of bi-Lipschitz constraints to preserve input space geometry in latent embeddings is well-motivated. (2) The developed model improved uncertainty quantification. The dual-pathway (global and local latent) design enhances the expressiveness and reliablity of uncertainty modeling. (3) It conducts extensive evaluation.The paper presents extensive empirical validation across synthetic and real-world tasks, including regression, classification, and OOD detection.

---

As for the weaknesses, I find the following concerns are waiting to address:

(1) Baselines are a bit outdated, I hardly see new after 2022. It would be necessary to include one of the following ones [1-5] published in NeurIPS/ICML/ICLR with others carefully discussed in the literature work.
(2) As the distance regularizer is the primary contribution in this work, it is necessary to include one ablation experiment to show Lipshitz term’s influence under varying scales.
(3) The limitation part should be added in the main paper.

---

Reference

[1] Nguyen, Tung, and Aditya Grover. "Transformer neural processes: Uncertainty-aware meta learning via sequence modeling." arXiv preprint arXiv:2207.04179 (2022).

[2] Wang, Qi, and Herke Van Hoof. "Learning expressive meta-representations with mixture of expert neural processes." Advances in neural information processing systems 35 (2022): 26242-26255.

[3] Tailor, Dharmesh, Mohammad Emtiyaz Khan, and Eric Nalisnick. "Exploiting inferential structure in neural processes." Uncertainty in Artificial Intelligence. PMLR, 2023.

[4] Wang, Qi, Marco Federici, and Herke van Hoof. "Bridge the Inference Gaps of Neural Processes via Expectation Maximization." The Eleventh International Conference on Learning Representations.

[5] Gondal, Muhammad Waleed, et al. "Function contrastive learning of transferable meta-representations." International Conference on Machine Learning. PMLR, 2021.

---

> ### Author Rebuttal · Authors · 2025-07-30
>
> We thank you for noting the clarity and thoroughness of the paper. Below, we address the weaknesses:
>
> **W1. Selection of Baselines and Related Work**
>
> We appreciate the references and will incorporate the following discussion of the works in the related work section.
>
> Transformer Neural Processes [1] uses autoregressive sequence modeling and replaces the traditional encoder-decoder architecture of NPs with a transformer-based approach that improves model expressiveness and ensures context invariance and target equivariance. [2] extends NPs by using multiple expert latent variables and assignment variables to capture multimodal distributions, addressing the expressiveness limitations of single global latent variables in standard NPs. [3] replaces NP's standard context aggregation with inference in probabilistic graphical models, treating context embeddings as neural sufficient statistics. The approach yields mixture and robust Bayesian Aggregation variants that show improved performance on out-of-distribution data and robustness to context corruption. [4] address the under-fitting problem in Neural Processes (NPs) from an optimization perspective. They identify that NPs struggle to learn accurate probability distributions during training as a key limitation, and propose a surrogate objective for the target log-likelihood within an expectation maximization framework to achieve improved performance. [5] propose a decoupled encoder-decoder approach where the encoder is trained with a contrastive objective to learn transferable meta-representations of functions.  This improved transferability across multiple downstream tasks and greater robustness to noise compared to end-to-end trained baselines. While these models focus on improving expressiveness or inference strategies, DNP takes a complementary approach by enforcing geometry-aware regularization through a bi-Lipschitz-constrained latent path. This inductive bias helps preserve input structure, leading to more coherent uncertainty estimates and stronger OOD performance.
>
> Below, you can find the metrics obtained for [3] ,referred to as NP-mBA on the 1D Synthetic regression task. DNP still achieves the best performance compared to the baselines.
>
>
>
> | Method         | RBF LL ↑         | RBF ECE ↓        | Matérn-5/2 LL ↑  | Matérn-5/2 ECE ↓ | Periodic LL ↑     | Periodic ECE ↓    |
> |----------------|------------------|------------------|------------------|------------------|-------------------|-------------------|
> | CNP            | 0.490 ± 0.006    | 0.122 ± 0.061    | 0.419 ± 0.004    | 0.088 ± 0.047    | 0.239 ± 0.001     | 0.120 ± 0.029     |
> | NP             | 0.480 ± 0.009    | 0.102 ± 0.082    | 0.318 ± 0.008    | 0.111 ± 0.053    | 0.228 ± 0.002     | 0.145 ± 0.047     |
> | ConvCNP        | 0.496 ± 0.007    | 0.118 ± 0.054    | 0.329 ± 0.001    | 0.098 ± 0.032    | 0.335 ± 0.002     | 0.106 ± 0.032     |
> | ConvNP         | 0.609 ± 0.005    | 0.190 ± 0.074    | 0.313 ± 0.002    | 0.137 ± 0.095    | 0.431 ± 0.003     | 0.117 ± 0.011     |
> | AttnNP         | 0.578 ± 0.003    | 0.103 ± 0.026    | 0.499 ± 0.009    | **0.078 ± 0.014**| 0.457 ± 0.001     | 0.098 ± 0.038     |
> | DSVNP          | 0.582 ± 0.002    | 0.101 ± 0.032    | 0.512 ± 0.006    | 0.101 ± 0.012    | 0.412 ± 0.002     | 0.092 ± 0.042     |
> | NP-mBA         | 0.521 ± 0.005    | 0.108 ± 0.012    | 0.324 ± 0.007    | 0.091 ± 0.021    | 0.375 ± 0.005     | 0.091 ± 0.010     |
> | **Ours**       | **0.696 ± 0.003**| **0.093 ± 0.054**| **0.620 ± 0.002**| 0.083 ± 0.010    | **0.635 ± 0.001** | **0.074 ± 0.022** |
>
> **W2. Ablation for the Bi-Lipschitz Term**
>
> In Appendix B.2, we already include ablation analyses over the lower and upper Lipschitz bounds ($\lambda_1$, $\lambda_2$) (Table 11), and over the regularization weight $\beta$ (Table 12). These results show that there is a trade-off between model expressiveness and regularization strength. When the bounds are too tight, the encoder becomes overly constrained, limiting the model's expressiveness. Conversely, when the bounds are too loose or the regularization weight is too small, the latent space can become distorted, leading to degraded uncertainty calibration.
>
> **W3. Including Limitations in the Main Paper**
>
> We discuss the limitations (scalability and Euclidean assumptions) and potential future directions to address them in Section 6 of the main paper. We will create a dedicated “Limitations” subsection and move the relevant content there.

---

> > ### Comment · Reviewer_HQey · 2025-08-02
> >
> > After reading the rebuttal, my concerns and questions are mostly addressed. And I updated the score accordingly.

---

### Official Review · Reviewer_n9Sq · 2025-07-02

**Clarity:** 3
**Significance:** 2
**Originality:** 3
**Rating:** 5
**Confidence:** 4

**Summary:**

The paper introduces *distance-informed neural processes* (DNP), which extends attentive NPs (AttnNPs) by encouraging the representations of the context and target points to maintain their relative distances from the input space. The authors achieve this by regularising the encoder through a bi-Lipschitz constraint, which encourages the singular values of the weight matrices to be within some predefined upper and lower bounds. The resulting model is trained through variational inference and evaluated on both regression and classification problems, outperforming baseline models on most tasks.

**Questions:**

**Suggestions**
1. As mentioned under weaknesses, I am not convinced that the motivation for introducing the bi-Lipschitz constraint (i.e., to maintain relative distances) is helpful. I would suggest focusing instead on the smoothness properties of the non-linear transformation, which, I realise, is to some extent a different way of saying the same, but it would be clear that you are not trying to learn an identity mapping (which Figure 1c seems to be and which would be counterproductive).

**Questions**
1. It seems to me that your approach to imposing the bi-Lipschitz constraint is readily applicable to other architectures, e.g., convolutions. Is this correctly understood? If so, did you test bi-Lipschitz ConvNP or similar?
2. I’m curious to understand the effect of the smoothness constraint on the model performance. As another ablation study, did you try to train the same architecture as the DNP but with some standard regularisation (e.g. weight decay) instead of the bi-Lipschitz constraint?
3. What do the errors in the tables refer to? They seem quite small.
4. Did you try to visualise samples from your model, e.g., for the 1D synthetic experiment? AttnNP can suffer from weird kinks in the samples, and I am curious to see if the bi-Lipschitz constraint had an effect here.

**Ethical Concerns:**

["NO or VERY MINOR ethics concerns only"]

**Final Justification:**

The authors have mostly addressed my questions and concerns, and while I still see the submission as an incremental contribution, I think it deserves to be accepted.

**Limitations:**

Yes.

**Paper Formatting Concerns:**

None noticed.

**Quality:**

3

**Strengths And Weaknesses:**

**Strengths**
1. The proposed model appears original and generally outperforms the baselines significantly.
2. The experimental section is extensive. In particular, the authors have performed several ablation studies to understand the effect of the introduced smoothness constraint.
3. The paper is well-written and well-structured.

**Weaknesses**
1. While the model seems original, its novelty is limited. The paper seems quite incremental, and given that adding the bi-Lipschitz constraint to the objective function is a fairly small change, I would have expected a more thorough analysis of the effect of the constraint on, e.g., training stability, sample quality, and other NP architectures, say, the ConvNP.
2. The description of the proposed model and its relation to other models from the literature could be improved, for instance with a side-by-side comparison of their (simplest possible) graphical models and/or key equations.
3. While the reference list is extensive, there are surprisingly few papers on neural processes or neural network smoothness cited. And even among those that are cited, it is not entirely clear how the current model positions itself in the literature.
4. I found the motivation for introducing the bi-Lipschitz constraint and Figure 1 confusing and a bit misleading. To me, it sounds like the constraint is imposed to try to enforce that the relative distances of the data are unchanged by the encoding, which Figure 1 seems to support, but that would essentially mean that the encoding is an identity map. Rather, the bi-Lipschitz constraint imposes smoothness in the encoding transformation, and I think this is a more helpful mental picture of what the proposed model does and why it should work.

---

> ### Author Rebuttal · Authors · 2025-07-30
>
> We thank you for the thoughtful and constructive feedback. Below we address the weaknesses and questions.
>
> **W1, Q1. Method Novelty and additional bi-Lipschitz analysis**
>
> We would like to clarify that our contribution lies in the novel integration of architectural and regularization components that, together, lead to significant empirical gains in uncertainty modeling. In particular, DNP is the first to introduce a **bi-Lipschitz-constrained local latent path** within the NP framework. This promotes the model’s capacity to capture local structure and produce more **calibrated uncertainty estimates**.
>
> Please note that we have included additional ablation analyses on the impact of bi-Lipschitz bounds and the trade of parameter between the ELBO and the bi-Lipschitz loss in the Appendix B.2 Tables 11 and 12 respectively. Our results show that there is a trade-off between model expressiveness and regularization strength.
>
> We will include the following additional results when applying bi-Lipschitz regularization to ConvCNP and ConvNP.
>
>
> | Method                  | RBF LL ↑         | RBF ECE ↓       | Matérn-5/2 LL ↑   | Matérn-5/2 ECE ↓ | Periodic LL ↑     | Periodic ECE ↓   |
> |-------------------------|------------------|------------------|-------------------|------------------|-------------------|------------------|
> | ConvCNP                 | 0.496 ± 0.007     | 0.118 ± 0.054     | 0.329 ± 0.001      | 0.098 ± 0.032     | 0.335 ± 0.002      | 0.106 ± 0.032     |
> | ConvNP                  | 0.609 ± 0.005     | 0.190 ± 0.074     | 0.313 ± 0.002      | 0.137 ± 0.095     | 0.431 ± 0.003      | 0.117 ± 0.011     |
> | ConvCNP + bi-Lipschitz  | 0.507 ± 0.012     | 0.086 ± 0.013     | 0.400 ± 0.014      | 0.081 ± 0.014     | 0.399 ± 0.004      | 0.082 ± 0.021     |
> | ConvNP + bi-Lipschitz   | 0.702 ± 0.008     | 0.073 ± 0.018     | 0.362 ± 0.016      | 0.093 ± 0.019     | 0.515 ± 0.003      | 0.098 ± 0.018     |
>
> The results indicate that adding the bi-Lipschitz constraint improves the log-likelihood and ECE scores of the base ConvCNP and ConvNP models.
>
> **W2. Comparative Model Descriptions**
>
> We currently provide textual comparisons of the baselines in Sec. 2 and 4. In the revised version, we will include the following table highlighting the architectural and probabilistic differences between the various baselines.
>
>
> | NP Model | Recognition Model | Generative Model | Prior Distribution | Latent Variable |
> |------------|-------------------|------------------|--------------------|------------------|
> | **CNP**    | $\mathbf{z}_C = f(\mathbf{x}_C, \mathbf{y}_C)$ | $p(\mathbf{y}_T \mid \mathbf{z}_C, \mathbf{x}_T)$ | - | Global (Deterministic) |
> | **NP**     | $q(\mathbf{z}_G \mid \mathbf{x}_T, \mathbf{y}_T)$ | $p(\mathbf{y}_T \mid \mathbf{z}_G, \mathbf{x}_T)$ | $p(\mathbf{z}_G \mid \mathbf{x}_C, \mathbf{y}_C)$ | Global (Stochastic) |
> | **ConvCNP**      | $\mathbf{z}_i = f(\mathbf{x}_C, \mathbf{y}_C, \mathbf{x}_i)$                        | $p(\mathbf{y}_i \mid \mathbf{z}_i)$                            | -                                                                                   | Local (Deterministic)           |
> | **ConvNP**       | $\mathbf{z}_i \sim f(\mathbf{x}_C, \mathbf{y}_C, \mathbf{x}_i)$             | $p(\mathbf{y}_i \mid \mathbf{z}_i)$   |  Implicit via f | Local (Stochastic) |
> | **AttnNP** | $q(\mathbf{z}_G \mid \mathbf{x}_T, \mathbf{y}_T)$, $\mathbf{z}_i = f(\mathbf{x}_C, \mathbf{y}_C, \mathbf{x}_i)$ | $p(\mathbf{y}_i \mid \mathbf{z}_G, \mathbf{z}_i, \mathbf{x}_i)$ | $p(\mathbf{z}_G \mid \mathbf{x}_C, \mathbf{y}_C)$ | Global (Stochastic) + Local (Deterministic) |
> | **DSVNP**  | $q(\mathbf{z}_G \mid \mathbf{x}_T, \mathbf{y}_T)$, $q(\mathbf{z}_i \mid \mathbf{z}_G, \mathbf{x}_i, \mathbf{y}_i)$ | $p(\mathbf{y}_i \mid \mathbf{z}_G, \mathbf{z}_i, \mathbf{x}_i)$ | $p(\mathbf{z}_G \mid \mathbf{x}_C, \mathbf{y}_C)$, $p(\mathbf{z}_i \mid \mathbf{z}_G, \mathbf{x}_i)$ | Global (Stochastic) + Local (Stochastic) |
> | **DNP (Ours)**    | $q(\mathbf{z}_G \mid \mathbf{x}_T, \mathbf{y}_T)$, $q(\mathbf{z}_i \mid \mathbf{x}_C, \mathbf{y}_C, \mathbf{x}_i)$ | $p(\mathbf{y}_i \mid \mathbf{z}_G, \mathbf{z}_i, \mathbf{x}_i)$ | $p(\mathbf{z}_G \mid \mathbf{x}_C, \mathbf{y}_C)$, $p(\mathbf{z}_i \mid \mathbf{x}_C, \mathbf{y}_C, \mathbf{x}_i)$ | Global (Stochastic) + Local (Stochastic) |
>
> Unlike AttnNP and DSVNP, which also use global and local latent variables, DNP incorporates a bi-Lipschitz constraint on the local latent path. This improves uncertainty calibration and OOD detection.
>
> **W3. Related Work and Literature Positioning**
>
> Our model extends the NP family by introducing a distance-aware inductive bias in the latent path, inspired by non-parametric modeling methods such as Gaussian Processes. The use of bi-Lipschitz regularization aligns with prior efforts to control neural network sensitivity and preserve structure [1, 2], but ours is, to our knowledge, the first to apply it within the NP setting to improve uncertainty calibration and OOD robustness. We will revise the related work section in the final version to better articulate this positioning, along with the description and positioning of the additional references suggested by Reviewer HQey.
>
> **W4 , S1. Motivation and Interpretation of Bi-Lipschitz Regularization and Figure 1**
>
> We thank the reviewer for pointing out this potential source of confusion. In Figure 1, we used a special case of our method where the bi-Lipschitz bounds were set to $\lambda_1$ = $\lambda_2$ = 1, resulting in an isometric mapping. This was done intentionally for visualization purposes, to clearly highlight the issues of over-sensitivity and feature collapse in the latent space, and to show how the bi-Lipschitz constraint mitigates them. However, we acknowledge that not stating this explicitly in the paper was an oversight, and we will clarify this setup and its illustrative intent in the text.
> That said, we agree that the main use of the bi-Lipschitz constraint is not to enforce exact distance preservation, but rather to promote smoothness and control sensitivity in the encoder. This interpretation provides a more general and helpful understanding of the regularizer’s role. In the figure, the encoder maps from a 2D input to a 2D latent space using a shallow network, which makes the transformation appear close to an identity. However, in all our experiments, we use relaxed bounds (e.g., $\lambda_1$ = 0.1, $\lambda_2$ = 1) and project to a latent space of dimension different from the input. In these scenarios, identity mappings are neither desirable nor possible.
>
> **Q2. Effect of smoothness constraint**
>
> Thank you for the thoughtful question. The ablation study in Table 6 investigates the effect of the smoothness constraint on DNP's performance by comparing several regularization strategies.
> We treat DNP without regularization as one extreme case where no explicit smoothness constraint is enforced. On the other extreme, we evaluate gradient penalty and orthogonal regularization, which strongly constrain all singular values to be close to 1, essentially enforcing a near-isometry. Both these scenarios degrade the model’s performance and uncertainty calibration.
> In contrast, spectral normalization only bounds the largest singular value, promoting Lipschitz continuity but allowing lower singular values to collapse. Our proposed bi-Lipschitz regularization lies between these extremes: it enforces both upper and lower bounds on singular values, promoting smooth yet expressive transformations by avoiding both over-sensitivity and feature collapse.
> We now include results for CIFAR10 vs CIFAR100 using the standard weight decay value of 1e-4.
>
> Accuracy =   91.92, ECE = 0.038, AUPR = 87.84
>
> Compared to the no-regularization case (ECE = 0.042, AUPR = 87.85), weight decay has a marginal improvement on uncertainty calibration. In contrast, bi-Lipschitz regularization significantly improves both ECE and AUPR.
>
> **Q3. Error in the table**
>
> The errors in our tables represent the standard deviation of the mean, computed across multiple evaluation tasks (10 runs). As the metrics are averaged over many data points and tasks, the resulting error bars are small and reflect the stability of the metric estimates, not predictive uncertainty. If I misunderstood the question, please let me know.
>
> **Q4. Visualization of sample**
>
> In our experiments, DNP's predictive samples were smooth, and we did not observe any “kinks”. Additionally, as shown in Figure 3, we observed that AttnNP occasionally produces excessively high uncertainty in certain regions, while simultaneously underestimating uncertainty in some OOD regions. This behavior is consistent with the qualitative findings reported by [3]. We hypothesize that both these issues in AttnNP stem from geometric distortions in the latent space. This is especially problematic because AttnNP relies directly on this latent space for obtaining the attention values during inference. In contrast, the smoothness property of the bi-Lipschitz constraint in DNP helps mitigate this. As a result, we obtain coherent samples and calibrated uncertainty estimates.
>
>
> [1] Jeremiah Liu, Zi Lin, Shreyas Padhy, Dustin Tran, Tania Bedrax Weiss, and Balaji Lakshminarayanan. Simple and principled uncertainty estimation with deterministic deep learning via distance awareness. Advances in neural information processing systems, 33:7498–7512, 2020
>
> [2] Joost Van Amersfoort, Lewis Smith, Andrew Jesson, Oscar Key, and Yarin Gal. On feature collapse and deep kernel learning for single forward pass uncertainty. arXiv preprint arXiv:2102.11409, 2021
>
> [3] Qi Wang and Herke Van Hoof. Doubly stochastic variational inference for neural processes with hierarchical latent variables. International Conference on Machine Learning, pages 10018–10028. PMLR, 2020.

---

> > ### Comment · Reviewer_n9Sq · 2025-08-02
> >
> > Thank you very much for your rebuttal, which cleared things up for me. I am happy to see the additional experiments, and the new model comparison table is very helpful. I am also happy to see that you will include a discussion of the papers mentioned by reviewer HQey, and I hope that you will include them as baselines for the experiments as well, if possible. I have increased my score accordingly.

---

### Official Review · Reviewer_C6NN · 2025-07-03

**Clarity:** 3
**Significance:** 3
**Originality:** 3
**Rating:** 4
**Confidence:** 4

**Summary:**

This paper introduces the Distance-informed Neural Process (DNP), a new member of the Neural Process family designed to improve uncertainty quantification and generalization. The authors identify a key weakness in standard NPs: their reliance on a single global latent variable often leads to poorly calibrated uncertainties and a failure to capture local data dependencies. To overcome this, DNP employs a dual-path architecture. A global path, similar to standard NPs, captures task-level uncertainty. The main contribution is a local latent path, which models input similarity explicitly. This is achieved by learning a distance-preserving latent space for the inputs, enforced through a novel bi-Lipschitz regularization on the encoder network. The similarity between points in this regularized latent space is then used to compute attention weights, allowing the model to make predictions that are informed by local context. Through extensive experiments on regression and classification tasks, the paper demonstrates that DNP achieves superior predictive performance, better-calibrated uncertainty estimates, and more effective out-of-distribution (OOD) detection compared to several state-of-the-art NP variants.

**Questions:**

1. **Connection to Classical Methods**: The core idea of using distance-based weights for local modeling is conceptually close to classical non-parametric methods like locally weighted regression (LWR) or kernel regression. Could you elaborate on the relationship between DNP and these methods? Specifically, what are the key advantages of your deep learning approach with bi-Lipschitz regularization compared to a modern variant of LWR or a deep kernel learning model that also aims to preserve local similarity?

2. **Scalability of the Local Path**: The O(NM) complexity of the cross-attention mechanism is a practical limitation for large context sets. You mention exploring sparse attention as future work. Have you considered or run preliminary experiments on simpler, more scalable approximations? For instance, one could attend only to the k-nearest neighbors in the latent space. How does performance degrade under such approximations? A brief analysis would provide valuable insight into the performance-efficiency trade-off.

**Ethical Concerns:**

["NO or VERY MINOR ethics concerns only"]

**Limitations:**

yes

**Quality:**

3

**Strengths And Weaknesses:**

Strengths
- The paper proposes a well-motivated and technically sound solution to a known limitation in NPs by integrating a distance-aware local latent path.
- The method's effectiveness is convincingly demonstrated through comprehensive experiments showing strong performance, especially in uncertainty calibration and OOD detection.
- The paper is exceptionally clear and well-written, with intuitive figures and a thorough appendix that aids understanding and reproducibility.

Weaknesses
- The work is an incremental advancement over existing NP variants, combining known concepts in a novel way rather than introducing a completely new paradigm.
- The model's O(NM) computational complexity from cross-attention presents a scalability challenge for tasks with large context sets.
- The proposed bi-Lipschitz regularization is limited to Euclidean spaces, which may hinder performance on data with non-Euclidean geometries.

---

> ### Author Rebuttal · Authors · 2025-07-30
>
> We would first of all like to thank you for the positive evaluation of the paper’s motivation, clarity and experimental rigor. We now address the weaknesses and the questions raised.
>
> **W1. Incremental Nature of the Contribution**
>
> While DNP builds upon existing Neural Process variants, we emphasize that our work introduces a **novel integration of architectural and regularization techniques** that together yield substantial empirical improvements in uncertainty modeling. We are the **first to incorporate a bi-Lipschitz-constrained local latent path** within the NP framework. This introduces an inductive bias that improves the model’s ability to capture local structure and produce well-calibrated uncertainty estimates. Empirically, DNP consistently outperforms the strong NP baselines such as AttnNP, ConvNP, and DSVNP across a variety of regression and classification tasks (Tables 1-5). These results indicate that DNP offers a principled and practically useful advancement in NP design.
>
> **W2, Q2. Scalability of Local Path**
>
> We evaluated the kNN-based sparse attention variant on the CIFAR10 (ID) vs. CIFAR100 (OOD) classification task. The results for 3 different values of k are as follows:
>
> | k   | Accuracy (%)     | ECE             | AUPR (%)        |
> |-----|------------------|------------------|------------------|
> | 5   | 87.34 $\pm$ 0.01     | 0.021 $\pm$ 0.002    | 87.34 $\pm$ 0.02     |
> | 10  | 88.19 $\pm$ 0.01     | 0.019 $\pm$ 0.003    | 87.92 $\pm$ 0.01     |
> | 20  | 89.05 $\pm$ 0.02     | 0.019 $\pm$ 0.001    | 89.08 $\pm$ 0.01     |
>
> These results indicate that increasing k generally improves model performance. However, even small values of k (e.g., k = 5) yield competitive results, suggesting that the model can achieve strong performance even with limited neighborhood information.
>
> **W3. Euclidean Limitation of Bi-Lipschitz Regularization**
>
> We agree that the current bi-Lipschitz constraint in DNP preserves Euclidean geometry in the latent space. We would like to clarify that our goal in this work is to take a principled first step toward introducing distance awareness into NPs, and to show that even simple distance preservation can significantly improve uncertainty calibration.
> Even though  this constraint may limit applicability in settings with inherently non-Euclidean data structures (e.g., graphs or manifolds), we note that our approach is already effective across a broad range of regression and classification tasks involving high-dimensional and structured inputs. Moving forward, we see this as a foundation for future extensions, including the use of task-specific or non-Euclidean similarity metrics as noted in Section 6. We believe this opens up a promising research direction for incorporating more general geometric priors into the latent variables.
>
> **Q1. Connection to Classical Methods**
>
> We really appreciate this thoughtful observation. Indeed, the bi-Lipchitz preservation in the local latent path in DNP is conceptually inspired by distance-based non-parametric methods. The classical methods, though excellent in uncertainty calibration, are computationally expensive and lack flexibility when dealing with high-dimensional inputs. The main advantage of our method lies in **scalability and generalization**. Modern methods like Deep Kernel Learning (DKL) combine neural feature extractors with Gaussian Processes (GPs), but they inherit the computational limitations of GPs: $O(N^3)$ training time, where $N$ is the number of training points. Even sparse approximations with inducing points reduce this only to $O(NM^2)$, where $M$ is the number of inducing points, which remains costly for large datasets. In contrast, NPs, including DNP avoids these costs by framing the problem within a meta-learning framework. Once trained, DNP requires only $O(N'M)$ computation, where $M$ is the number of context points and $N'$ is the number of target points. Moreover, DNP can generalize across tasks without retraining. This is unlike DKL which typically requires retraining or fine-tuning per task. In summary, DNP thus offers a scalable and flexible alternative that preserves the inductive bias of distance-based classical non-parametric methods while being better suited for modern large-scale, task-distributed learning scenarios.

---

### Note · Authors · 2025-08-12

We thank the reviewers for their constructive feedback and for recognizing our work’s **originality**, **technical soundness**, **clarity**, and **empirical rigor**. Our Distance-informed Neural Process (DNP) introduces the first integration of a bi-Lipschitz-constrained local latent path within the NP framework. This enforces a geometry-aware inductive bias, helping the model preserve meaningful input relationships and better capture local structure in its predictions. This leads to consistent gains in uncertainty calibration, predictive performance, and OOD detection across a range of regression and classification tasks.


Following the reviewer feedbacks, we will include the following updates to the paper.

1. Include all additional experimental results: sparse attention using k-NN (C6NN), ConvNP and ConvCNP with bi-Lipschitz (n9Sq), NP-mBA comparison (HQey), CIFAR100 latency scores (jh3q).

2. Expand the related work to incorporate recent NP variants (HQey) and improve literature positioning (n9Sq).

3. Add the baseline comparison table summarizing architectural/probabilistic differences (n9Sq).

4. Clarify Figure 1’s setup and illustrative role (n9Sq).

5. Explain the observed underestimation of DNP’s uncertainty in the qualitative analysis (jh3q).

6. Include limitations as a separate section (HQey).

These updates are already prepared and will be fully incorporated. We believe that the clarifications and new results provided during the rebuttal fully address the reviewers’ concerns and further demonstrate DNP as a principled and broadly applicable approach for improving uncertainty estimation in Neural Processes.

---

### Decision · Program_Chairs · 2025-09-17

**Decision:**

Accept (poster)

**Comment:**

The paper introduces the Distance-informed Neural Process (DNP) designed to improve uncertainty quantification and generalization in NPs. In addition to a global latent path as implemented in standard NPs the model employs a local latent path, which models input similarity explicitly. This is implemented by distance-preserving bi-Lipschitz regularization on the encoder network used to compute attention weights to local context.

All reviewers agreed that this paper is well motivated, technically sound and well written.
There were some concerns regarding the incremental nature of the work, and more specific concerns about comparisons and ablations, however the authors have addressed most of them in the rebuttal.